# The effect of artificial selection on phenotypic plasticity in maize

Joseph L. Gage et al.[#]

Remarkable productivity has been achieved in crop species through artificial selection and adaptation to modern agronomic practices. Whether intensive selection has changed the ability of improved cultivars to maintain high productivity across variable environments is unknown. Understanding the genetic control of phenotypic plasticity and genotype by environment (G × E) interaction will enhance crop performance predictions across diverse environments. Here we use data generated from the Genomes to Fields (G2F) Maize G × E project to assess the effect of selection on G × E variation and characterize polymorphisms associated with plasticity. Genomic regions putatively selected during modern temperate maize breeding explain less variability for yield G × E than unselected regions, indicating that improvement by breeding may have reduced G × E of modern temperate cultivars. Trends in genomic position of variants associated with stability reveal fewer genic associations and enrichment of variants 0–5000 base pairs upstream of genes, hypothetically due to control of plasticity by short-range regulatory elements.

#A full list of authors and their affliations appears at the end of the paper

The expression of an individual's phenotype is a function of its genotype (G), the environment experienced during its lifetime (E), and the complex relationship established by the differential sensitivity of certain genotypes to specific environmental influences throughout their lifetime. This variable plastic response[1] is referred to as genotype by environment interaction (G × E). Plants have evolved unique mechanisms to respond to variable environments because they are fixed in a specific location and cannot seek shelter or alter their environment. These plastic responses include changes in physiology, metabolism, growth, and development in response to biotic and abiotic stresses[2,3]. Natural populations with a greater capacity for phenotypic plasticity have been shown to have greater fitness than populations that are less able to respond to their environment[4], supporting an evolutionary advantage conferred by plastic responses. Conversely, plasticity can also confer an evolutionary disadvantage in novel environments when there has not yet been selection on genetic variation for plasticity[5]. Plasticity is heritable, and therefore can be intentionally selected for or against in manmade populations[6]. Artificial selection in the form of modern crop breeding has yielded remarkably productive cultivars that are stable across diverse conditions, but it is not clear whether phenotypic plasticity is among the traits that have been selected for or against.

Variable plasticity of genotypes across differing environments can be quantified as G × E. Proposed mechanisms of the genetic basis for G × E include overdominance, pleiotropy, epistasis, linkage, and epigenetic causes[7]. Heritable plasticity may contribute to a population's success in novel habitats, but may also contribute to divergence of populations in different environments[8]. If a population is introduced to and becomes successful in a novel habitat, but is subsequently restricted to that habitat by other forces, alleles that contributed to plastic adaptation in the new environment could trend towards fixation in the absence of gene flow from other populations[9–12]. We hypothesize that as these loci under selection for fitness in the new environment trend toward fixation, their ability to confer plasticity is also subsequently reduced.

Expression of abiotic stress responses by plants is a complicated process, thought to involve (among other mechanisms) differential response of gene regulatory networks[13–15]. Genes involved in stress response can be classified as either coding for proteins that directly protect against stress or as coding for products that regulate downstream gene expression[16]. DNA polymorphisms that lie within gene promoter regions have been associated with G × E variation for flowering time in Arabidopsis[17]. The complex nature of G × E, along with results from attempts to map genes responsible for G × E variation[11,17–19]

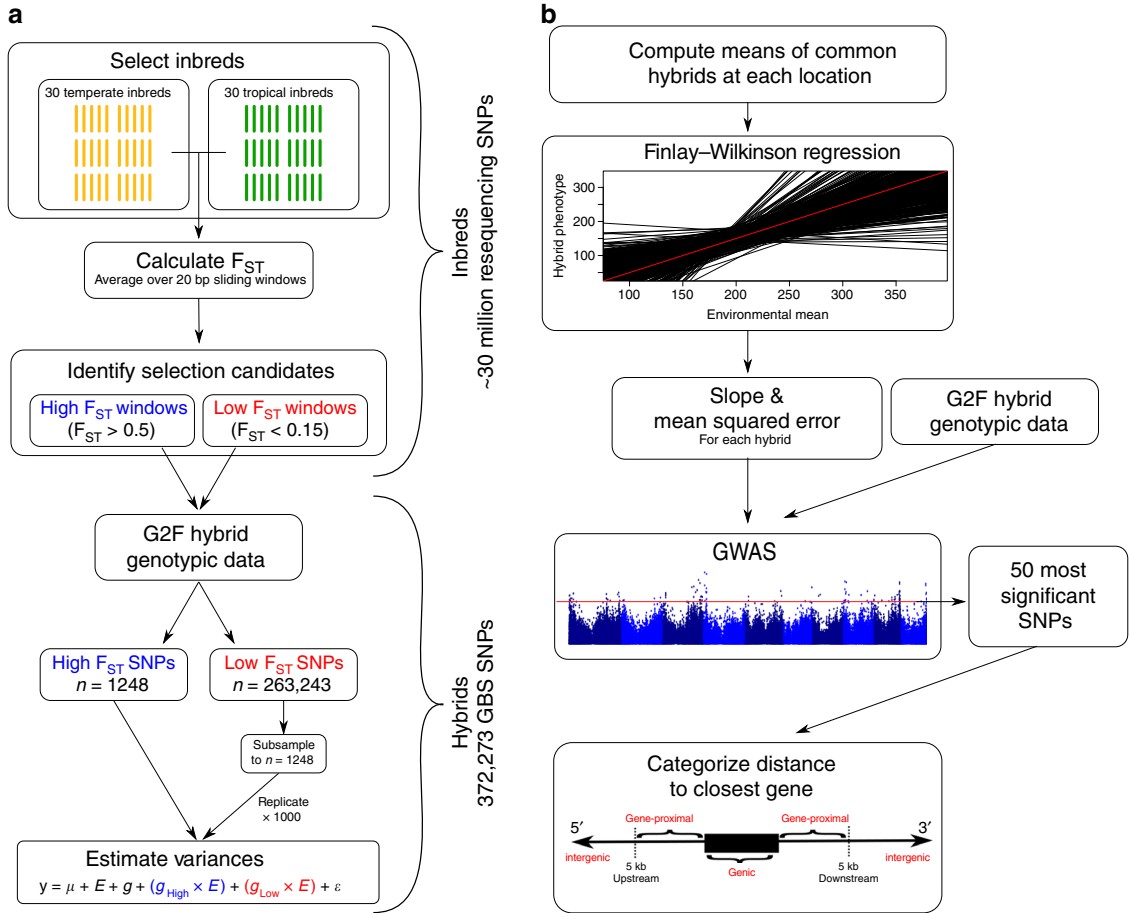

**Fig. 1** Flowchart of experimental analyses. To investigate how putatively selected regions influence variation for G × E (**a**), 30 temperate and 30 tropical inbreds were used to calculate $F_{ST}$ in 20 base pair sliding windows across the genome. Windows with mean $F_{ST} > 0.5$ were categorized as "high" $F_{ST}$, and windows with mean $F_{ST} < 0.15$ were categorized as "low" $F_{ST}$. SNPs from the Maize G × E project hybrids that were within high or low $F_{ST}$ windows were categorized as high or low $F_{ST}$ SNPs, and used to estimate G × E variances attributable to high and low $F_{ST}$ regions of the genome. To investigate location of variants associated with G × E (**b**), hybrid phenotypes were regressed on the means of common hybrids at each location. The slope and mean squared errors from each hybrid's regression were used as response variables in GWAS, and the 50 most significant SNPs from each GWAS were evaluated for their position relative to the nearest gene

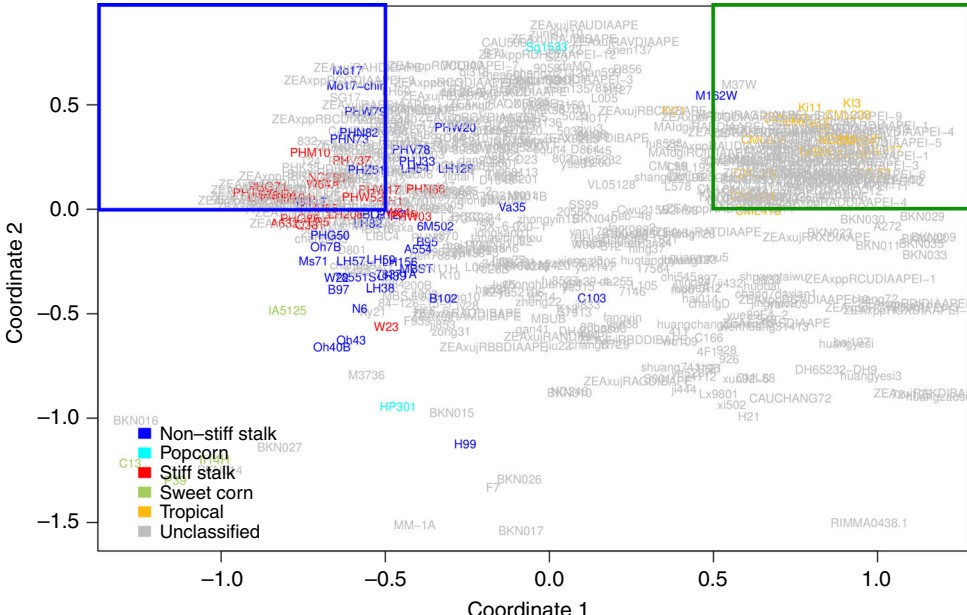

**Fig. 2** MDS of genotypes used to selected 60 extreme individuals. Unique inbred individuals ($n = 916$) from the HapMap 3.1 visualized by multi-dimensional scaling (MDS). The temperate materials are bound by the blue box (coordinate 1 < −0.5, coordinate 2 > 0), and the tropical materials are bound by the green box (coordinate 1 > .5, coordinate 2 > 0). Two sets of 30 individuals were chosen from each box based on pedigree, genetic distance from others in the group (identity by state < 0.95), and quantity of missing SNP data

suggest that genetic control of G × E is highly polygenic. One of the hypothesized reasons that some mapping studies only explain small amounts of variation is because highly complex traits are controlled by numerous polymorphisms with small effects[19,20]. The evidence for regulatory regions controlling stress responses, combined with mapping evidence that points to G × E variation being controlled by many small-effect loci, lead us to hypothesize that G × E variation is disproportionately controlled by numerous regulatory mechanisms.

Studies that discuss the genetic control and modulation of plasticity and G × E variation have been conducted either in the context of natural populations or model species evaluated in contrasting conditions within controlled environments. Surveys of existing natural populations, long used by ecologists, do not have the structure of replicated, designed experiments that cultivated species can provide. On the other hand, evaluations in controlled environments can impose extreme conditions leading to overestimation of the variation found in natural conditions[21]. There is a deficit of large-scale field experiments that study species in the context of defined, variable growing conditions. Crop species grown in typical field production environments can provide such an experimental structure.

The Maize (*Zea mays* L.) G × E Project, launched in 2014, is a part of the Genomes to Fields (G2F) initiative (www.genomes2fields.org). This project has measured phenotypes of maize hybrids across a geographically and climatically diverse transect of the North American maize growing landscape.

Maize, as both a model species and a crop grown worldwide, is an ideal candidate for replicated, field-based studies of G × E variation across a wide range of environments. Maize was domesticated in southwestern Mexico[22–24] and has since been adapted to be productive in a variety of habitats and growing conditions. As a major crop that has undergone widespread adaptation to novel environments, maize affords us an opportunity to investigate the genetic basis for G × E variation in the context of productivity traits (e.g., yield) as well as phenological traits (e.g., flowering time and plant height), which display great variability.

This study leverages data generated by the Maize G × E project to study the relationships between allelic variation and G × E. Specifically, we investigate how loci showing evidence of differential selection affect G × E and where single nucleotide polymorphisms (SNPs) associated with G × E tend to be located in the genome. We test the following hypotheses: (1) genomic regions that have experienced changes in allele frequency due to selection for productivity in temperate conditions explain less G × E variation than regions in which allele frequency was unaffected by selection; and (2) G × E variation is disproportionately controlled by regulatory mechanisms.

To test the first hypothesis, we use high quality resequencing data in groups of temperate and tropical inbred maize lines to identify regions that show high divergence in allele frequency between the two groups. We then use SNP data from hybrids grown as part of the Maize G × E project to estimate G × E variance explained by those divergent regions relative to a set of SNPs that show little to no divergence (Fig. 1a), and provide evidence that putatively selected genomic regions show reduced contribution to G × E for grain yield.

To address the second postulation, we perform a Finlay–Wilkinson regression[25] on the hybrids grown for the Maize G × E project. We use the slope and mean squared error (MSE) parameters from those regressions as the response variables in genome-wide association studies (GWAS). We then examine the genomic location of polymorphisms associated with G × E in relation to nearby genes, providing evidence for enrichment of associations in regulatory regions of the genome (Fig. 1b).

## Results

**Partitioning phenotypic variance.** The Maize G × E project grew 858 unique hybrids in a modified split-plot design at 21 locations across the North America. The hybrids were derived from 8 inbred pools crossed by up to five male testers. Phenotypic data were collected for 11 morphological and agronomic traits in 12,678 field plots.

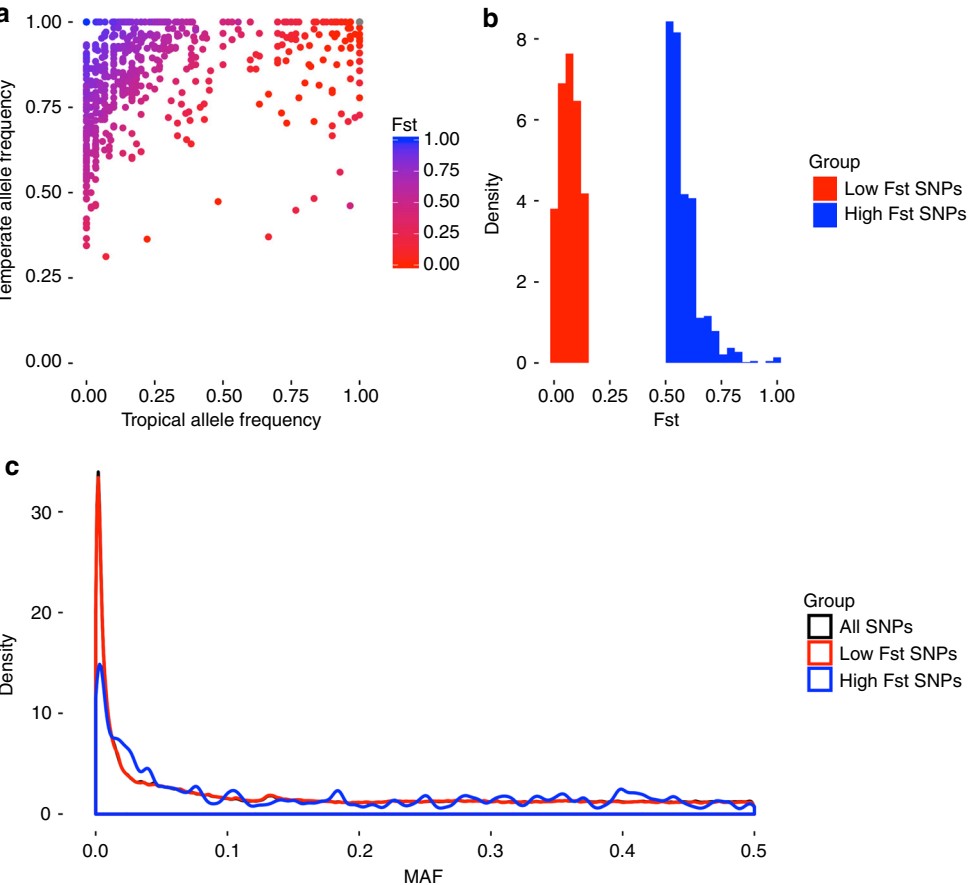

**Fig. 3** Comparisons of high and low $F_{ST}$ SNPs. **a** Allele frequencies within the 30 temperate and 30 tropical inbred lines from Hapmap 3.1 for 736 high $F_{ST}$ SNPs that overlap between Hapmap 3.1 and the G × E hybrid lines. Some SNPs with $F_{ST} < 0.5$ were designated as high $F_{ST}$ because they lie in a window with mean $F_{ST} > 0.5$. **b** Histograms of $F_{ST}$ distributions of 1248 high $F_{ST}$ SNPs and 263,243 low $F_{ST}$ SNPs from the G × E hybrid data set. $F_{ST}$ values represent means of 20-SNP windows. **c** Distributions of the minor allele frequencies (MAF) in the G × E hybrid data set for 1248 high $F_{ST}$ SNPs, 263,243 low $F_{ST}$ SNPs, and the entire set of 372,273 polymorphic SNPs

A wide range of responses were observed for the phenotypic traits evaluated in this study (Supplementary Fig. 1). Raw phenotypic values averaged within location had ranges between locations of 23 and 25 days for days to anthesis and silk, respectively; 78 and 48 centimeters for plant and ear height, respectively; 107 bushels per acre for yield; 17 and 18 pounds for plot and test weight, respectively; 36 plants for stand; and 22% for moisture. These ranges are attributable to both genotypic differences between hybrids, as well as environmental and experimental effects. We partitioned phenotypic variance for each trait in accordance with the field experimental design to determine the proportion of variance attributable to each design parameter. A wide array of environmental conditions were recorded across the 21 locations included in this evaluation (e.g., rainfall and temperature; Supplementary Figs. 2, 3). As expected, given the variety of climatic conditions, the largest proportion of the observed variance was consistently attributed to the environment term. Environmental variance consisted of between 42% (for ear height) and 74% (for test weight) of the total variance (Supplementary Fig. 4). Variance attributable to differences between hybrids, estimated by the hybrid-within-set term of the model, comprised between 4% (test weight) and 15% (ear height) of the total variance. G × E was modeled as hybrid by environment within set, and contributed between 1% (days to silk) and 6% (yield) of the overall variance. Residual error component accounted for between 4% (days to anthesis) and 25% (ear height).

**G × E variance explained by high and low $F_{ST}$ regions**. Two groups of inbred lines were identified from the entire HapMap 3.1[26] collection that represented extreme selection differentiation. In Fig. 2, relative distance between individuals indicates relative genetic distance, enabling visualization of divergence between modern, temperate maize lines and tropical materials primarily along the first coordinate. Based on differences in allele frequency between the two groups, two contrasting sets of candidate SNPs were chosen that exhibited evidence of potentially having been either selected or not selected during modern breeding in temperate environments. Candidate SNPs were chosen by assessing allele frequency changes with $F_{ST}$. The 1248 high $F_{ST}$ SNPs were chosen from genomic regions with mean $F_{ST} > 0.5$ and had an overall mean $F_{ST}$ of 0.58, while the 263,243 low $F_{ST}$ SNPs were chosen from windows with mean $F_{ST} < 0.15$ and had an overall mean $F_{ST}$ of 0.07. By choosing 0.5 as a cutoff, the pool of high $F_{ST}$ SNPs still includes SNPs with intermediate allele frequencies, rather than being constrained to SNPs that are nearly fixed in one or both populations (Fig. 3). Mean $F_{ST}$ of the low $F_{ST}$ candidate SNPs was low enough to provide a large contrast between the high and low $F_{ST}$ groups, despite the mean of the high $F_{ST}$ SNPs being 0.58. Because SNPs were designated as high $F_{ST}$ based on 20-SNP means, some individual high $F_{ST}$ SNPs did not have an $F_{ST}$ value greater than 0.5. Based on the 736 SNPs that are present in both Hapmap 3.1 and the hybrid line genotypes, we estimate that 28.7% of the SNPs designated as high $F_{ST}$ actually have individual $F_{ST}$ values less than 0.5 but lie in a window with mean

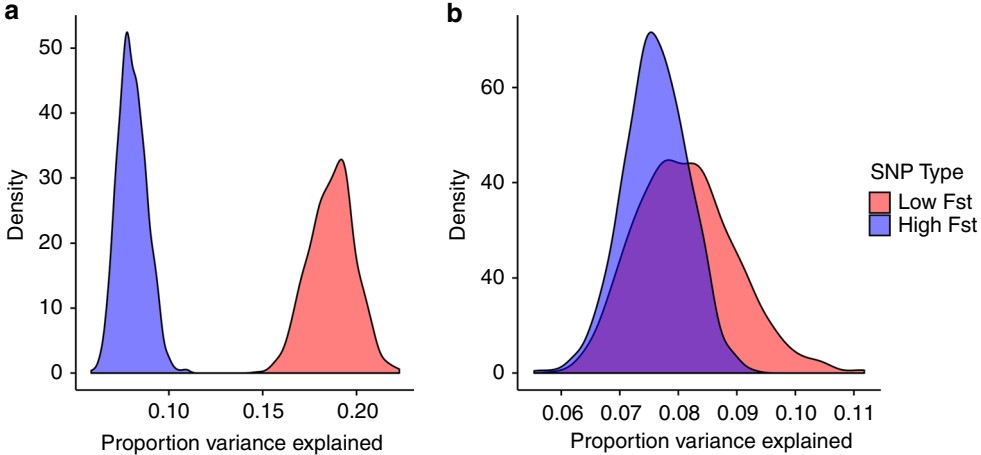

**Fig. 4** Empirical distribution of estimated variance components for high and low $F_{ST}$ G × E interaction. Distributions of G × E variance attributable to high and low $F_{ST}$ SNPs for grain yield (**a**) and plant height (**b**) from 1000 replicated model fittings. 1248 high $F_{ST}$ SNPs were included, while each model fitting used a subsample of 1248 low $F_{ST}$ SNPs chosen randomly from the full set of 263,243 low $F_{ST}$ SNPs. Proportion variance explained represents non-environmental model variance, i.e., was calculated using only genotype, G × E for high and low $F_{ST}$ SNPs, and residual variances

$F_{ST} > 0.5$. Regions with high $F_{ST}$ could occur due to four scenarios: (1) they were selected in both temperate and tropical material; (2) they were selected in temperate material and experienced little or no selection in tropical material; (3) they experienced little or no selection in temperate material and selection in tropical material; or (4) they are the result of random genetic drift. Our conclusions are contingent on the assumption that the majority of high $F_{ST}$ regions were selected on in the temperate material, i.e., that the third and fourth categories are not disproportionately large. Analysis of nucleotide diversity in the high and low $F_{ST}$ regions revealed most of the high $F_{ST}$ regions had low nucleotide diversity in both temperate and tropical materials (Supplementary Fig. 5). The presence of low nucleotide diversity in both populations (rather than one or the other) across many of the high $F_{ST}$ regions provides further evidence for divergent selection in those regions. The median nucleotide diversity in high $F_{ST}$ regions for both temperate and tropical lines (0.0020 and 0.0026, respectively) is similar to median nucleotide diversity previously reported for known selection candidates in maize (0.0021)[27]. We used a variance components approach (adapted from Gusev et al.[28]) to calculate the phenotypic variance of 552 hybrids attributable to SNP-by-environment interaction of SNPs with differential $F_{ST}$. Due to concern that different allele frequencies between high and low $F_{ST}$ SNPs within the set of hybrids could affect the variance estimates, we compared the minor allele frequency (MAF) distributions for the high $F_{ST}$ and low $F_{ST}$ SNPs but observed no major differences (Fig. 3). High and low $F_{ST}$ SNPs were also compared for distance to the nearest gene, proportion of imputed sites, LD among SNPs, and distance among SNPs. There were no major differences between the sets of SNPs for distance to the nearest gene or proportion of imputed sites. High $F_{ST}$ SNPs generally had higher among-SNP LD and were closer to each other than the low $F_{ST}$ SNPs, as would be expected for loci clustered on selected genomic regions. Because a lack of genetic variance could cause decreased G × E variance, we also calculated the genetic variance separately for the high and low $F_{ST}$ SNPs. We observed reduced genetic variance attributable to high $F_{ST}$ SNPs compared to low $F_{ST}$ SNPs for both grain yield and plant height. Genetic variance captured by low $F_{ST}$ SNPs was 21.0% for grain yield and 33.5% for plant height, but high $F_{ST}$ genetic effects still accounted for 11.2% (grain yield) and 20.8% (plant height) of the non-

environmental variance (Supplementary Fig. 6), demonstrating that a non-trivial proportion of genetic variance exists within the high $F_{ST}$ SNPs.

We analyzed variance components for plant height and grain yield to evaluate whether interactions between high $F_{ST}$ regions and environment show differences in the amount of phenotypic variability explained. We included SNP-by-environment interaction terms for both high (putatively selected) and low (putatively unselected) $F_{ST}$ SNPs in our random effects model, allowing us to partition the phenotypic variance that was attributable to environment, genetic effects, and G × E of presumably selected and unselected genomic regions (Supplementary Fig. 7).

For grain yield, more variance was explained by the interaction between low $F_{ST}$ SNPs ($n = 263,243$, subsampled to $n = 1248$ for each iteration of model fitting) and environment than by the interaction between high $F_{ST}$ SNPs ($n = 1248$) and environment (Fig. 4). For plant height, on the other hand, we observe little difference in variance explained by interaction between environment and low $F_{ST}$ vs. high $F_{ST}$ SNPs. Across all 1000 iterations of the grain yield model, high $F_{ST}$ SNPs captured less G × E variance than low $F_{ST}$ SNPs, while for plant height the high $F_{ST}$ SNPs captured less G × E variance than low $F_{ST}$ SNPs in only 63% of model iterations. For grain yield, the interaction between low $F_{ST}$ SNPs and environments controlled more than 2.3 times as much variance as the high $F_{ST}$ SNPs by environment term. Setting aside the estimate of environmental variance, leaving only variance components attributable to genetic effects, we calculated the proportion of remaining variance attributable to high and low $F_{ST}$ SNPs. For grain yield, the variance explained ranged from 5.9 to 11.0% with a mean of 8.1% for high $F_{ST}$ SNPs and from 14.7 to 22.2% with mean of 18.7% for low $F_{ST}$ SNPs. For plant height, the variance captured by high $F_{ST}$ SNPs ranged from 5.5 to 9.3% with a mean of 7.6%, while low $F_{ST}$ SNPs controlled from 5.9 to 11.2% of the variance, with a mean of 8.1%. These results indicate that regions that show evidence of differential selection between temperate and tropical germplasm explain less G × E variance for grain yield than those that do not, suggesting that selection for high productivity during temperate maize breeding has reduced the G × E variation for that trait in modern germplasm. A different pattern is observed for plant height, likely because the same selection effort has not explicitly focused on changing plant height.

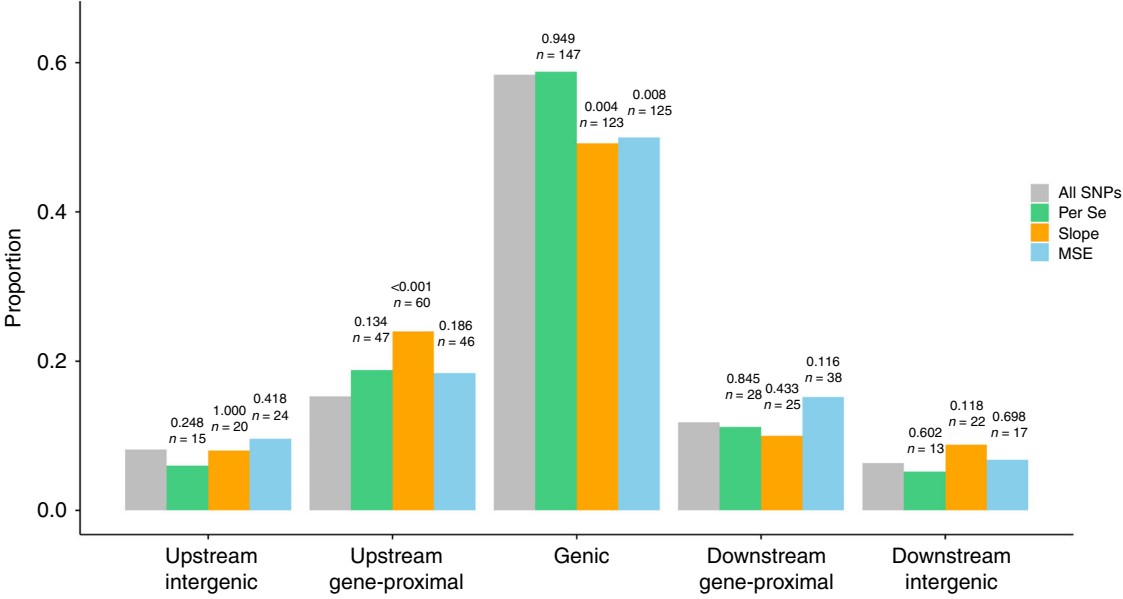

**Fig. 5** Patterns of functional variation and classification of SNPs based on their distance to the nearest gene model. Proportions of 250 slope(type II stability)-associated, 250 MSE(type III stability)-associated, and 250 phenotype per-se-associated SNPs in genic, gene-proximal (0–5000 base pairs from nearest gene), and intergenic (>5000 base pairs from nearest gene) regions compared to a null distribution of proportions derived from all 413,796 SNPs. Text above each bar indicates sample sizes for each bin and two tailed $p$-values from an exact binomial test for the null hypothesis of underlying proportion equal to the null distribution. For $\alpha = .05$, the Bonferroni multiple testing threshold is $\alpha = .01$

**Classification of variants associated with G × E.** To determine whether significant SNPs controlling G × E are primarily genic or non-genic, we first quantified plasticity of each hybrid using a method similar to that originally described by Finlay and Wilkinson[25]. We started by performing simple linear regression of each hybrid's location-specific phenotypes against an environmental gradient, which was calculated as the mean phenotype at each location of the common hybrids that were grown across at least 20 environments. We did this separately for plant height, ear height, days to silk and anthesis, and grain yield. Using the slope and MSE parameters resulting from these regressions, we were able to assess two measures of plastic response for each hybrid. These measures of plasticity are also referred to as type II and type III stability[29]. Lines that are said to display type II stability have a response to changing environments that is parallel to the average response for that environmental gradient. Genotypes with a slope near one have changes in performance parallel to the checks, and are therefore determined to be type II stable. Type III stability is characterized by having little variation around a line regressing performance on ordered environmental indices. In the case of this experiment, hybrid lines with low MSE are considered to be type III stable. By using slope and MSE as the response variables in GWAS, we were able to identify genomic loci that are associated with different types of stability. Additionally, we performed GWAS on the traits per se using the hybrid line best linear unbiased predictions (BLUPs) derived from the experimental design random effects model.

To categorize the variants identified from GWAS, we assigned variants into classes based on proximity to the closest annotated gene model. We pooled the top 50 most significant SNPs from GWAS of each trait's slope, resulting in 250 slope-associated SNPs for the five traits considered. MSE-associated and trait-per-se-associated SNPs were pooled in the same manner. There was no systematic co-localization of slope-, MSE- or trait-per-se-associated SNPs (Supplementary Fig. 8). We then computed the distance from the slope-, MSE-, and trait-per-se-associated SNPs to the nearest annotated gene model, allowing us to determine whether the associated SNPs were genic or non-genic. The non-genic SNPs were determined to be either upstream or downstream of the nearest gene based on annotated gene orientation, and they were classified as gene proximal if the closest gene was <5000 bp away. Although the SNPs that were identified by GWAS are not necessarily causative for changes in stability or traits per se, they may be in linkage with and therefore physically proximal to causative variants that were not genotyped.

When we compared the distributions of distance to the nearest gene for non-genic slope- and MSE-associated SNPs with the distance distribution of all SNPs (Fig. 5), we observe enrichment ($p = 0.0003$, two-sided exact binomial test) for slope-associated SNPs in the upstream gene-proximal region relative to what would be expected from the null distribution. A Bonferroni (5 tests per parameter) corrected type I error rate of 0.05 is 0.01 ($0.05/5 = 0.01$). The upstream gene-proximal region corresponds with the typical location of promoters and short-range regulatory elements. The rest of the distance distribution for non-genic slope-associated SNPs, and the entire distribution of non-genic MSE-associated SNPs, are similar to the distribution formed from all SNPs. Both slope- and MSE-associated genic SNPs were reduced ($p = 0.004$ and 0.008, respectively; two-sided exact binomial test) relative to the all-SNP distribution. The decrease in genic SNPs contrasts with the results of Wallace et al.[30], who found a strong enrichment of genic SNPs in GWAS hits of phenotypic traits per se. GWAS hits for the traits per se in this study did not differ from the all-SNP distribution in any category.

These results provide evidence that plastic response in maize is disproportionately associated with non-genic regions of the genome and that in the case of type II stability, variation is attributable to the upstream gene-proximal region where short-range regulatory elements are frequently located. We do not see a similar pattern for type III stability, indicating that different types of plastic response might be under different types of regulation.

## Discussion

The results presented here provide evidence to support the hypotheses that (1) genomic regions selected for high

productivity in temperate environments show reduced contribution to G × E variation, supported by the difference in grain yield G × E variance explained by high and low $F_{ST}$ regions; and (2) that G × E variation is disproportionately controlled by regulatory mechanisms, supported by enrichment of upstream gene-proximal variants associated with stability.

Our results provide evidence that regions of the maize genome which were presumably selected during modern temperate maize breeding contribute less to grain yield G × E variance than genomic regions that do not appear to have been selected. These results rely heavily on the assumption that the $F_{ST}$ statistic can reliably identify genomic regions that have been differentially selected; high $F_{ST}$ values can also occur due to random genetic drift, which we assume here to have had less of an effect at the high $F_{ST}$ regions than selective forces. Analysis of nucleotide diversity among genomic regions with high and low $F_{ST}$ revealed that the majority of high $F_{ST}$ regions had low nucleotide diversity in both temperate and tropical materials, supporting the idea that genetic differentiation in high $F_{ST}$ regions is due to divergent selection. Additionally, while $F_{ST}$ between temperate and tropical materials has been calculated with as few as 16 and 11 individuals in each group[27], even the sample size of 30 and 30 used in this study may result in considerable noise surrounding estimates of $F_{ST}$. Finally, different sets of lines used for calculating $F_{ST}$ may also result in the identification of differing selection candidate loci, but the approach of defining groups based on genetic differences across the axis separating temperate and tropical lines is the most compelling given that maize originated in tropical regions and adapted to temperate locations.

We interpret these results as evidence that genomic regions that originally contributed to the adaptation of maize to temperate North American growing conditions may now limit the ability of modern North American temperate germplasm to adapt to different environments. Through continued breeding, the modern germplasm pool will produce future hybrids that are adapted to new environmental conditions, yet may no longer contain alleles that once conferred plasticity. Limited plastic response can be either beneficial or antagonistic in cultivar development. Frequently, G × E is the result of biotic or abiotic stress susceptibilities that are exposed in particular environments. Breeders' approach to G × E has traditionally been to either reduce or exploit the phenomenon[31]. To reduce G × E, breeders attempt to select lines that perform consistently across a target population of environments (i.e., display stability). For example, breeders try to produce cultivars that perform reliably despite year-to-year fluctuations in weather patterns. In the case where limited plastic response confers stability, the low G × E contribution of selected regions may have a desirable effect by enabling germplasm to perform predictably across environments. Modern temperate maize as a whole was heavily selected for stable grain yield but not explicitly for stable plant height. The large difference in G × E explained by high and low $F_{ST}$ SNPs for grain yield, compared to the small difference for plant height, reflects this systematic difference in how the two traits were selected. Under the framework of G × E being the result of susceptibilities exposed in specific environments, the low G × E and higher stability conferred by high $F_{ST}$ regions may be a sign of effective selection by maize breeders against alleles that are deleterious in modern temperate breeding programs. When exploiting G × E, breeders attempt to select lines that perform particularly well in specific environments. For example, most modern maize is bred to mature more or less quickly depending on where it will be grown. If genetic potential for plastic response is limited, it decreases breeders' ability to identify cultivars that truly excel in specific environments. That is, reduced G × E contribution of selected regions may have the undesirable effect of constraining performance potential when cultivars are bred for specific locales. Based on these findings, we are unable to know whether this pattern of decreased G × E variation attributable to selected regions is unique to the adaptation of North American maize germplasm, or whether similar trends would be seen in other highly selected populations of maize (e.g., maize adapted to tropical environments) or in different species.

Our second hypothesis, that phenotypic plasticity is disproportionately controlled by regulatory regions, is also supported by the results of this study. Because the experimental design was unbalanced and hybrids were assigned to locations based on expected maturity, there is likely to be some degree of genotype-environment correlation. This is an inherent limitation of experiments that measure crop productivity in relevant field conditions. Results for this experiment rely on the assumption that parameters of the Finlay–Wilkinson regressions are good estimators of their true values despite measuring hybrids in a non-random subset of environments.

A previous study[30] found enrichment for associations between phenotypes and variants in both gene-proximal and genic regions. The associations were mapped using phenotypes per se—that is, measurements of physical traits. Our results from mapping the stability of cultivars for various traits across a range of environments revealed a similar pattern of gene-proximal associations with type II stability, but differ in that we observed a depletion of genic associations with both type II and type III stability. Our mapping of phenotypes per se did not reveal any deviations from the null distribution. This contrasts with the aforementioned study, which observed enrichment for genic and gene-proximal variants associated with traits per se using inbred maize lines. Our study was conducted with hybrids, in which major allele effects may be more likely to be complemented or otherwise buffered, complicating additive mapping efforts. The deviation of stability-associated SNPs from the expected distribution established by the full set of SNPs provides evidence that we are seeing some true mapping associations rather than purely false positives or noise. We hypothesized that phenotypic plasticity is largely controlled by regulatory elements rather than genes per se, and the enrichment of gene-proximal variants associated with type II stability provides support for this hypothesis. The enrichment of upstream associations in gene-proximal regions precludes the possibility that we had an enrichment of gene-proximal variants only because they were in linkage disequilibrium with genic causal mutations. If that were the case, we would have expected to see enrichment for both upstream and downstream gene-proximal variants. Observing enrichment of upstream gene-proximal variants for type II stability but not type III stability means we are seeing gene-proximal association with differences in the linear response of cultivars across environments, but not with variability around that linear response. Due to the way type II and type III stability were calculated (slope and MSE), estimates of type II stability provide some degree of smoothing across environments while estimates of type III stability are more sensitive to stochastic differences between environments. As a result, it is unclear whether the lack of upstream gene-proximal enrichment for type III associated variants is attributable to differential control of type II and type III stability or just experimental noise. The observed decrease in genic associations with both type II and type III associated variants was not an effect that we predicted, but may be cautiously interpreted as further evidence that stability is modulated even less by structural genes per se than anticipated. These findings could be further validated with the addition of denser SNP data, more phenotype data for an array of traits, and a larger number of phenotyped and genotyped individuals.

The G2F Maize G × E project is an ongoing experiment that is accumulating phenotypic data across years and locations. The analyses presented here are an example of the project's utility, which will only increase as data continues to grow. This experiment fits a yet unfilled niche in the study of phenotypic plasticity and adaptation; specifically, that of a large, multi-environmental replicated experiment that has some of the attractive features of both natural population and controlled-environment studies.

## Methods

**Germplasm and plant growth conditions**. A total of 858 unique maize hybrids were tested in 21 environments across 14 states in the United States and one province in Canada for a total of 12,678 field plots in the summer of 2014 as a part of the G2F initiative. Each of the 21 environments grew a set of 250 hybrids in two field replications. The environments ranged from latitudes between 30.54° and 44.07° and longitudes between −101.99° and −75.20°. For more details about specific agronomic practice and growing conditions for each location, please refer to the metadata at https://doi.org/10.7946/P2V888. Female parents of hybrids were classified into eight pools based on genetic background. Briefly, those pools were: (1) Recombinant inbred lines (RILs) from the Intermated B73/Mo17 population (IBM)[32]; (2) RILs from the nested association mapping (NAM)[33] population involving B73/Oh43; (3) RILs from the NAM population involving B73/Ki3; (4) Public and expired plant variety protection (PVP) lines belonging to the Iodent group; (5) Public and expired PVP lines belonging to the Stiff Stalk group; (6) Public and expired PVP lines belonging to the Lancaster group; (7) Public lines originating from the Texas A&M AgriLife corn breeding programs; (8) RILs developed by the University of Wisconsin's biomass breeding program. The first maize inbred sequenced[34], B73, is a parent of pools 1–3 and a founding member of the Stiff Stalk group represented by pool 5. Pools 4 and 6 represent the Iodent and Lancaster groups which are commonly crossed to Stiff Stalk materials in public and private breeding programs. Pool 7 represents temperate and exotic germplasm selected for adaptation to Texas[35], while pool 8 contains RILS derived from diverse parents showing segregation for various biomass related traits. Pools 1 through 8 were crossed with up to five male testers (PB80, LH195, CG102, LH198, and LH185), and each pool-by-tester family was designated as a "set" (Supplementary Table 1). In addition to the sets created by the crosses described above, there were two additional sets: the first comprised of single locally adapted (in some cases commercial) check hybrids chosen by each principal investigator for their location, and the other comprised of a common set of hybrids grown in all locations. Sets were assigned to specific locations based on expected maturity with the exception of the set of common check hybrids, which were grown in all locations, and the locally adapted checks, which were grown only in their individual locations.

**Field experimental design**. The experiment followed a modified form of a split-plot design with individual sets as the whole-plot factor arranged in a randomized complete block design and hybrids as the subplot factor. The design differed from a classical split-plot because the subplot factor (hybrid) was nested within the whole-plot factor (set). This design is also referred to as a sets-in-replicates design. Two complete replicates of each hybrid were planted at each location; within each replicate, each set was grown in a block, and block order was randomized within replicate. Hybrid order was randomized within each whole-plot block. The locally adapted hybrid check selected by each investigator at each location was incorporated into each block within each replicate. Weather data were collected at each location (Supplementary Note 1).

**Phenotypic data**. Eleven morphological and agronomic traits were measured for all hybrids and across all locations. Methods for their measurement were standardized project-wide. A detailed description of phenotyping guidelines is available at https://doi.org/10.7946/P2V888. Days to anthesis was measured as the number of days between planting and half the plot exhibiting anther exertion on more than half of the main tassel spike. Days to silking was assessed as the number of days between planting and half the plot showing silk emergence. Ear height was the distance from the ground to the uppermost ear bearing node. Plant height was measured as the distance from the ground to the ligule of the uppermost leaf. Plot weight was the weight in grams of the shelled grain collected in each plot, and test weight (a measure of grain density) was recorded as pounds per bushel. Root lodging and stalk lodging were recorded respectively as the number of plants leaning more than 15 degrees from vertical and as the number of plants with broken stalks between the ground and the primary ear node. Stand count was recorded as the number of plants per plot at harvest. Grain moisture was measured as the percent water content in the grain at the time of grain harvest. Grain yield in bushels per acre assumed a 56 pounds per bushel conversion, 15.5% grain moisture, and used plot area measured without the alley. The calculation for grain yield was grain yield = (plot weight)×(1−0.01×moisture) × (area$^{-1}$) ×920.5401. Ear height and plant height were measured on one to five representative plants per plot depending on the location while all other measurements were representative of the entire plot. Full phenotypic data can be found at https://doi.org/10.7946/P2V888.

**Experimental design random effects model**. As detailed in the field experimental design section, hybrids were classified into sets based on the female pool and the male tester. Hybrid genotypes were grown in a modified split-plot design. To calculate the variance attributable to each element of the field experimental design, we modeled each phenotype as $y = E + R(E) + S + E \times S + R \times S(E) + L(S) + L \times E(S) + e$, where $E$ represents the environmental effect; $R(E)$ is the effect of replication nested within environment; $S$ is the set effect; $E \times S$ is the interaction term of environmental and set effects; $R \times S(E)$ is the interaction term of replication by set, nested within environment; $L(S)$ is the hybrid line effect nested within set; $L \times E(S)$ is the interaction term of hybrid line by environment, nested within set; and $e$ is the error term. Models were fit in R[36] using the *lmer()* function in the lme4 package[37]. Variance component estimates were expressed as a percentage of the total variance. Predictions of hybrid effects were recorded as the best linear unbiased predictions (BLUPs) for hybrid line nested within set.

**Genotypic data**. A set of 336 inbred lines were used to generate the hybrid sets tested in the 2014 experiment. Sequencing data for 232 of the inbred lines used in this evaluation were downloaded from the ZeaGBSv2.7 Panzea release (http://www.panzea.org/#!genotypes/cctl). DNA for the remaining inbreds was extracted and genotyped using genotype-by-sequencing (GBS) following the protocol described by Elshire et al.[38] at 96 plex. Genotypes were called using the Tassel5-GBS Production Pipeline with the ZeaGBSv2.7 Production TagsOnPhysicalMap(TOPM) file that was built using information about ~32,000 additional *Zea* samples[39] (AllZeaGBSv2.7_ProdTOPM_20130605.topm.h5, available at panzea.org). Imputation was performed with FILLIN[40] using the available set of maize donor haplotypes with 8k windows (AllZeaGBSv2.7impV5_AnonDonors8k.tar.gz, available at panzea.org). FILLIN has been shown to have an imputation accuracy of 0.996 on temperate inbred materials representative of the germplasm used in this study. All GBS samples used are listed in Supplementary Data 1. Available GBS data can be found at https://doi.org/10.7946/P2V888.

Synthetic hybrid genotypes for 624 hybrids were generated based on genotypes of parental inbreds. The subset of hybrids for which genotypes were calculated was based on availability and quality of parental genotypes, not deliberately chosen. Parental genotypes were coded as the number of major alleles at each locus (0, 1, or 2) and the hybrid genotype for each hybrid at each locus was computed as the mean of its two parents at that same locus.

**G × E variation explained by high and low F$_{ST}$ regions**. A set of 30 inbred lines which have undergone selection for high productivity in temperate growing conditions and 30 inbred lines selected for productivity in tropical climates were chosen to use for identification of genomic regions that are candidates for having undergone differential selection for growth in temperate conditions (Supplementary Table 2). Overlapping SNPs between ZeaGBS 2.7[39] and Hapmap 3.1[26] (341,048 SNPs) were used to perform a Multidimensional Scaling (MDS) analysis on the 916 *Zea* accessions described in Hapmap 3.1, following the procedure described by Romay et al.[41] Based on the location of the inbreds that were part of Hapmap 2[42], inbreds with coordinates <−0.5 on the first coordinate and above 0 on the second coordinate were classified as temperate selected while inbreds with coordinates > 0.5 on the first coordinate and greater than 0 on the second coordinate were classified as tropical selected (Fig. 2). Thirty individuals were chosen from each group (Supplementary Table 2) based on pedigree, genetic distance from other individuals of the group (identity by state <0.95[41]), and missing SNP data. Thirty individuals from each group was the sample size that best balanced the amount of missing data between temperate and tropical lines. Previous publications have computed F$_{ST}$ between tropical and temperate materials with as few as 16 and 11 individuals per group (respectively)[27]. F$_{ST}$[43] between the two groups was calculated for each SNP in Hapmap 3.1 using VCFtools[44]. The unweighted average was calculated to determine an F$_{ST}$ value for every 20-SNP interval. From the F$_{ST}$ results, 1248 SNPs in the hybrid lines were identified as present in regions that are more probable to have been subject to selection (windows with F$_{ST}$ values greater than 0.5), while 263,243 SNPs in the hybrid lines were chosen as present in regions that are unlikely to have been selected (windows with F$_{ST}$ values <0.15). Per-site pairwise nucleotide diversity was assessed in the high and low F$_{ST}$ regions within the temperate and tropical inbreds to assess evidence of divergent selection vs. directional selection within individual subpopulations. Nucleotide diversity calculations were performed with the Hapmap 3.1 sequencing data using SAMtools[45] and ANGSD[46]. Because the high and low F$_{ST}$ SNP groups were chosen based on mean values of 20-SNP windows, some of the SNPs that were included in the high F$_{ST}$ group do not have an individual F$_{ST}$ greater than 0.5. The 736 high F$_{ST}$ SNPs that overlap between Hapmap 3.1 and the hybrid line genotypes were used to evaluate allele frequencies between the temperate and tropical groups, as well as F$_{ST}$ values at individual SNPs (Fig. 3). Minor allele frequencies in the hybrid lines of the high F$_{ST}$ SNPs, low F$_{ST}$ SNPs, and entire SNP set were calculated as $\min(\frac{\mu_m}{2}, 1 - \frac{\mu_m}{2})$, where $\mu_m$ is the mean at marker $m$. Minor allele frequency distributions were compared visually, and no major differences between the distributions were noted (Fig. 3). Despite inbred imputation, 16% of 372,273 SNPs in the hybrid genotypic data were still missing, ranging from 0 to 56% missing on a per-SNP basis and from 3 to 56% missing on a per-hybrid basis. Missing hybrid genotypes for each marker $m$ were imputed by weighted random draws from the genotypes present at $m$, where the weights correspond to the genotype frequencies

of $m$. To calculate empirical allele frequency based imputation accuracy, we performed 10,000 iterations of masking a single known SNP and comparing it to its imputed value, with an empirical imputation accuracy of 85.6%.

We calculated G × E variance explained by SNPs with high and low $F_{ST}$ values using a method similar to the variance components approach described by Gusev et al.[28], but structured to evaluate interactions between the environments and specific loci rather than heritability estimations of functional categories[47]. The model describes the response of the $i$th hybrid in the $j$th environment as follows:

$$y_{ij} = \mu + E_j + g_i + (g_L E)_{ij} + (g_H E)_{ij} + e_{ij},$$

where $\mu$ is the overall mean; $E_j$ $(j = 1,...,J)$ denotes the random effect of the $j$th environment such that $E_j \overset{iid}{\sim} N(0, \sigma_E^2)$ with $\sigma_E^2$ as the variance of the environments; $g_i = \sum_{m=1}^{p} x_{im} b_m$ is a linear combination between $p$ marker covariates $x_{im}$ ($m = 1,..,p$) and their correspondent marker effects $b_m$ such that $g = \{g_i\} \sim N(0, G\sigma_g^2)$, where $G$ is the genomic relationships matrix (GRM) computed using all 227,287 polymorphic markers with minor allele frequency >0.05, and $\sigma_g^2$ is the genomic variance; $(g_L E)_{ij}$ represents the interaction between each SNP with low $F_{ST}$ and each environment such that $g_L E = \{(g_L E)_{ij}\} \sim N(0, (Z_g G_L Z_g')^\circ (Z_E Z_E')\sigma_{g_L E}^2)$, where $Z_g$ and $Z_E$ are the incidence matrices for genotypes and environments, $\sigma_{g_L E}^2$ is the associated variance parameter and 'o' stands for Hadamard or Schur (element-by-element) product between two matrices; similarly $(g_H E)_{ij}$ represents a random effect of the interaction between SNPs with high $F_{ST}$ and the environments with $g_H E = \{(g_H E)_{ij}\} \sim N(0, (Z_g G_H Z_g')^\circ (Z_E Z_E')\sigma_{g_H E}^2)$ and $\sigma_{g_H E}^2$ acting as the variance component. $G_H$ was a GRM computed using the 1248 SNPs whose $F_{ST}$ values were above 0.5 and $G_L$ was a GRM computed using random samples of 1248 SNPs from the low $F_{ST}$ set of 263,243 SNPs. This model was fit 1000 times with random subsets of the low $F_{ST}$ SNPs, and the calculated variance components were recorded. Models were fit using the BGLR[48] package in R[36]. Residuals across all model fittings followed a distribution approaching normality, and heuristic assessment of equal variance between environments[49] was satisfied. Therefore, common transformations of phenotypic data were not explored.

Because presence of G × E variance is dependent on the presence of both genetic and environmental variances, we tested for the presence of genetic variance attributable to high and low $F_{ST}$ SNPs using the same model as described above, but with the $g_i$ term split into $(g_H)_i$ and $(g_L)_i$ where $(g_H)_i$ was calculated using the 1248 high $F_{ST}$ SNPs, and $(g_L)_i$ was calculated using 1248 SNPs randomly subset from the 263,243 low $F_{ST}$ SNPs. The model was fit 1000 times for both grain yield and plant height and the calculated variance components were recorded.

The hypothesis being tested assumes that hybrids used in this field evaluation are representative of only temperate selected germplasm. A small number of hybrids had inbred line Ki3 as a parent, which is of tropical origin and as such could contain alleles that are not representative of germplasm selected for growth in temperate conditions. The variance decomposition analysis described above was run with hybrid lines containing Ki3 parentage both included and excluded, with no differences observed. With the hybrid lines containing Ki3 parentage removed from the data set, 552 unique hybrids were included in each model fitting.

**Classification of variants associated with G × E.** Hybrid stability was calculated using a method similar to the Finlay–Wilkinson regression[25]. Environmental means were calculated using 21 check hybrids that were grown in at least 20 of 21 environments. Ear height, plant height, number of days to silk and anthesis, and yield values for each hybrid were regressed on the respective environmental means by simple linear regression: $y_{ij} = \beta_0 + \beta_1 x_j + e_{ij}$, where $y_{ij}$ is the phenotype of replicate $i$ in environment $j$, $x_j$ is the mean of the checks in environment $j$, and $e_{ij}$ is a random error term. The deviation from a slope of one (i.e., $\beta_1 - 1$, representative of deviation from the mean response of the checks and hereafter referred to simply as slope) and mean squared error (MSE) for each hybrid's regression were recorded. Hybrids with less than six recorded observations or with observations recorded in less than four environments for a particular trait were excluded from further analysis.

BLUPs for the traits per se were calculated as the hybrid line within set effect from fitting the experimental design random effects model. Slope, MSE, and traits per se were used as response variables in a genome-wide associate study (GWAS) with the synthetic hybrid genotypic data. Synthetic hybrid SNPs were filtered to 413,796 SNPs with <80% missing data, a mean of 20% missing, and 95% of SNPs having less than 61% missing. GWAS was performed using the software GAPIT[50] (Supplementary Figs. 9–11), with minor allele frequency threshold of 0.5%, kinship calculated by the VanRaden method[51] using only individuals for which the response variable was present, and default parameters otherwise.

The 50 SNPs with the lowest $p$-values from each GWAS for slope were pooled. If any of the top 50 SNPs were within 5 kilobases of each other and in LD ($r^2 > 0.5$), only the most significant SNP was retained. LD was calculated using PLINK v1.9[52]. Results from each GWAS for MSE and the traits per se were pooled in the same manner. Base pair (bp) distances from the pooled SNPs to the closest gene were calculated in a manner similar to that described by Wallace et al.[30], but rather than calculate the absolute distance to the nearest gene we also calculated whether each SNP was upstream or downstream (5′ or 3′ based on annotated gene orientation in the B73 AGPv2 reference genome (ftp.gramene.org/pub/gramene/maizesequence. org/release-5b/filtered-set/ZmB73_5b_FGS.gff.gz). A null distribution of distance to the closest gene was calculated using all 421,142 SNPs in the hybrid genotype

data set. We chose 50 SNPs per trait/parameter combination because it closely represented the proportion of total SNPs used in the Wallace et al.[30] study. For the null, slope, MSE, and trait per se distances, SNPs were categorized as either upstream or downstream and as genic (within a gene), gene-proximal (1–5000 bp to closest gene), or intergenic (>5000 bp to closest gene). Tests for enrichment or reduction of slope- or MSE-associated SNPs in each position category were performed against the null distribution using a two-sided exact binomial test.

**Code availability.** Scripts for modeling variance attributable to high and low $F_{ST}$ regions can be found in Supplementary Software 1. Scripts for nucleotide diversity, stability analysis, GWAS, and distance to the nearest gene can be found at https://github.com/joegage/GxE_scripts.

**Data availability.** Hybrid phenotypic data, inbred genotypic data, weather data, metadata, and readme files are publicly available at https://doi.org/10.7946/P2V888. All relevant data are available from the authors upon request.

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

## Acknowledgements

The authors would like to thank the entire G2F Consortium for their help with this study and extend a hearty acknowledgment to Tim Beissinger, Bret Payseur, and Jeff Ross-Ibarra for their sound advice. Ms. Lisa Coffey for assistance organizing and conducting field trials at Iowa State University and Dustin Eilert and Marina Borsecnik for assistance organizing and conducting field trials at the University of Wisconsin, Madison. This project was supported by the National Research Initiative for Agriculture and Food Research Initiative Competitive Grants Program grant no. #2012-67013-19460 from the USDA National Institute of Food and Agriculture, USDA Hatch program funds to multiple PIs in this project, NSF Plant Genome Research Project #1238014, the USDA-ARS, the Ontario Ministry of Agriculture, Food, and Rural Affairs, the Iowa Corn Promotion Board, the Nebraska Corn Board, the Minnesota Corn Research and Promotion Council, the Illinois Corn Marketing Board, and the National Corn Growers Association.

## Author contributions

J.L.G. wrote the manuscript and performed stability and SNP distance analysis. D.J. performed high/low $F_{ST}$ SNP G × E analysis. C.R. did GBS SNP calling and $F_{ST}$ analysis. A.L., S.K., and E.S.B. contributed ideas for analysis and experimental design. Members of the G2F Consortium selected germplasm, designed experiments, phenotyped plant materials and compiled and curated phenotypic and weather data sets. N.d.L. recognized the need for, conceived of, and organized the experiment.

## Additional information

**Competing interests:** The authors declare no competing financial interests.

Joseph L. Gage [1], Diego Jarquin[2], Cinta Romay [3], Aaron Lorenz[4], Edward S. Buckler [5,3], Shawn Kaeppler[1], Naser Alkhalifah[6,7,1], Martin Bohn [8], Darwin A. Campbell[6,7], Jode Edwards[9], David Ertl[10], Sherry Flint-Garcia[11], Jack Gardiner[12], Byron Good[13], Candice N. Hirsch[4], Jim Holland [14], David C. Hooker[15], Joseph Knoll[16], Judith Kolkman[17], Greg Kruger[18], Nick Lauter[9], Carolyn J. Lawrence-Dill[6,7], Elizabeth Lee[13], Jonathan Lynch [19], Seth C. Murray[20], Rebecca Nelson[21,17], Jane Petzoldt[1], Torbert Rocheford[22], James Schnable[2], Patrick S. Schnable [7], Brian Scully[23], Margaret Smith[21], Nathan M. Springer[24], Srikant Srinivasan[25],

Renee Walton[6,7], Teclemariam Weldekidan[26], Randall J. Wisser[26], Wenwei Xu[27], Jianming Yu[7] & Natalia de Leon[1]

[1]Department of Agronomy, University of Wisconsin-Madison, Madison, WI 53706, USA. [2]Department of Agronomy and Horticulture, University of Nebraska-Lincoln, Lincoln, NE 68583, USA. [3]Institute for Genomic Diversity, Cornell University, Ithaca, NY 14853, USA. [4]Department of Agronomy and Plant Genetics, University of Minnesota-St Paul, St Paul, MN 55108, USA. [5]USDA-ARS Plant, Soil, and Nutrition Research Unit, Cornell University, Ithaca, NY 14853, USA. [6]Department of Genetics, Development and Cell Biology, Iowa State University, Ames, IA 50011, USA. [7]Department of Agronomy, Iowa State University, Ames, IA 50011, USA. [8]Department of Crop Sciences, University of Illinois at Urban-Champaign, Urbana, IL 61801, USA. [9]USDA-ARS Corn Insects and Crop Genetics Research Unit, Iowa State University, Ames, IA 50011, USA. [10]Iowa Corn Promotion Board, 5505 NW 88th Street, Johnston, IA 50131, USA. [11]USDA-ARS Plant Genetics Research Unit, University of Missouri, Columbia, MO 65211, USA. [12]Division of Animal Sciences, University of Missouri–Columbia, Columbia, MO 65203, USA. [13]Department of Plant Agriculture, University of Guelph, Guelph, ON, Canada N1G 2W1. [14]USDA-ARS Plant Science Research Unit, North Carolina State University, Raleigh, NC 27695, USA. [15]Department of Plant Agriculture, University of Guelph-Ridgetown Campus, Ridgetown, ON, Canada N0P 2C0. [16]USDA-ARS Crop Genetics and Breeding Research Unit, Tifton, GA 31793, USA. [17]Plant Pathology and Plant-Microbe Biology Section, School of Integrative Plant Science, Cornell University, Ithaca, NY 14853, USA. [18]West Central Research and Extension Center, University of Nebraska-Lincoln, North Platte, NE 69101, USA. [19]Department of Plant Science, Penn State University, University Park, Penn, PA 16802, USA. [20]Department of Soil and Crop Sciences, Texas A&M University, College Station, TX 77843, USA. [21]Plant Breeding and Genetics Section, School of Integrative Plant Science, Cornell University, Ithaca, NY 14853, USA. [22]Department of Agronomy, Purdue University, West Lafayette, IN 47907, USA. [23]USDA-ARS U.S. Horticultural Research Laboratory, Fort Pierce, FL 34945, USA. [24]Department of Plant and Microbial Biology, University of Minnesota, St. Paul, MN 55108, USA. [25]School of Computing and EE, Indian Institute of Technology Mandi, Kamand, Himachal Pradesh 175005, India. [26]Department of Plant and Soil Sciences, University of Delaware, Newark, DE 19716, USA. [27]Texas A&M AgriLife Research, Texas A&M University, Lubbock, TX 79403, USA

