## [Peer Review File · Nature Communications]

Reviewers' comments:

Reviewer #1 (Remarks to the Author):

This paper deals with important questions about the genes that control plastic responses of plants to their environments. The authors conclude that alleles that were selected based on yield in temperate conditions (high F_{st} SNPs) differ from unselected SNPs (low F_{st}) for their level of genotype-environment interaction. The analyses indicate a significant reduction in GxE due to selected SNPs. This result is biologically plausible, interesting, and potentially important. Nevertheless, I have some substantial concerns.

1) In order to argue that selection explains the plasticity differences between these two classes of polymorphisms, it is important to examine whether the high F_{st} and low F_{st} SNPs differ in ways that might be confounded with F_{st} . For example, do these two groups differ in distance from genes, distance from each other, LD among SNPs, proportions of imputed sites, or other confounding factors?

2) I am skeptical about some of the methodologies used in this study, especially the reliance on SNPs that have very high missing rates, and using trait information from genotypes with 80% missing data. (See Data Quality, below.)

3) In a large and polymorphic genome, identifying high and low F_{st} SNPs based on groups of 30 inbred individuals seems like a rather small sample size. Furthermore, these 30 genotypes are not fully independent, given that their identity can be as much as 94%. I presume there were constraints in your data set which kept you from using more individuals (otherwise, one would definitely prefer more genotypes). This sounds like a serious concern -- what are the counter arguments?

4) Assignment of sets to locations was based on expected maturity (line 198), hence there is a genotype-environment correlation. What is the magnitude of this correlation, and how does this influence these genetic outcomes?

5) The sequence and details of analyses is logical, but there is a lot going on, and it is not easy to follow. At the end of the introduction, please provide a brief road-map to your approach.

Data quality:

I think the filtering thresholds for including hybrids across environments are much too lax (pages 15 and 16). Across 21 environments, you are only excluding hybrids with fewer than four data points, or fewer than six environments. From this you cannot get a reliable slope, nor can you be certain that you have sampled the range of environmental space, or whether differential slopes might simply be due to sampling different parts of the gradient. For the genotypes used, did you check whether the regressions deviate from linearity? (An interaction of genotype by quadratic effects of the gradient?)

Using SNPs that might have as much as 79% missing data strikes me as excessive (line 315). Please provide numbers and distributions so that we can have a clear understanding of the extent of missing SNP data (across SNPs and across genotypes).

Please provide information on the success of imputation. In particular, for SNPs that were called with high confidence, what is the success rate when you hide this information and impute these SNPs?

For linkage disequilibrium calculations (line 360), what proportion of the data have non-missing data for both SNPs? What is the mean number of non-missing SNP pairs that was used for linkage disequilibrium calculations?

Minor points:

For those of us who don't work on maize, it would be helpful to have more background about the different populations, their histories, and how they differ.

It is not clear that these eight groups should be described as heterotic groups (line 185). What is the evidence that they are heterotic? Also, heterosis is a characteristic of a *cross* between different parents, how can they be called heterotic unless you specify a comparison between groups?

“We observed reduced genetic variance attributable to high F_{st} SNPs compared to low F_{st} SNPs for both grain yield and plant height, but high F_{st} genetic effects still accounted for 11.2% (grain yield)” -- We also need to know how this 11.2% figure compares to the genetic variance of grain yield explained by low F_{st} SNPs.

The meaning of "set" (a pool-by-tester family) is not clear from the written description. I understand that a set has a group of female parents consisting of a given "heterotic" pool. Does this set have a single male tester (such as PB80), or does a given set have five male parents (PB80 through LH185)? In the total experiment, what is the number of sets, and how many full

sib-ships are there? What is the total sample size of the experiment analyzed here?

The different results between type II and type III stability (lines 505-509) may not reflect a biological difference. Estimation of slope for type II stability provides a smoothing affect across different environments, whereas stochastic differences among different environments will inflate type III estimates of variability. Thus, experimental noise may explain the lack of relationship between regulatory elements and type III stability.

“populations with a greater capacity for phenotypic plasticity have been shown to have greater fitness than populations that are less able to respond to their environment” (line 121) -- On the other hand, other examples find that plasticity may be non-adaptive (e.g., Ghalambor, et al, 2015, Non-adaptive plasticity potentiates rapid adaptive evolution of gene expression in nature. Nature 525: 372-375).

For the MDS, What proportion of the variance is explained by the first two axes? Is this sufficient to identify the two groups of 30 genotypes?

You suggest (line 536) that selection against deleterious alleles may explain the higher stability of selected SNPs. Does this imply that they are unconditionally deleterious? This is very different than constitutively deleterious alleles.

What factors might explain the difference in results between Wallace et al. vs. the current study?

Line 353: What is a “non-monomorphic hybrid genotype”? Here, is genotype synonymous with SNP?

Reviewer #2 (Remarks to the Author):

Gage and colleagues present results of a large-scale collaborative field trial, population genomic analysis, and GWAS mapping in maize to explore the impact of domestication on gene by environment interaction (GxE). GxE plays an important role in plant growth and performance and the study frames the question of how artificial selection during domestication affects standing GxE variation for agronomic traits. The implications of the study are broad, important and potentially provide insight into the mechanisms leading to the evolution of broadly stable versus regionally adapted cultivars.

The projects leverages an exciting new collaboration, the G2F project, wherein a common set of

maize hybrids and checks are grown at many locations across North America. The material was phenotyped for a core set of agronomic traits ranging from flowering time to grain yield and the plant material genotyped using a combination of resequencing, GBS, and imputation. These data are analyzed using statistical genetic approaches to partition sources of variation into E, G, GxE, and design factors. This multi-location trial is novel in that the plant material was picked to broadly represent maize diversity, tests effects in hybrids, and can be linked by association to SNP variation.

One of the main goals for the study is to ask whether genomic regions under selection during domestication affects variance in GxE. The authors attempt to identify these regions by comparing SNP variation in a chosen set of 30 temperate and 30 tropical adapted maize inbreds. This material would have been the result of strong selection during breeding programs and the authors aim to identify the regions which responded to selection by identifying outliers in the relative differentiation metric F_{st} .

F_{st} measures the degree of among group versus within group allelic variation. It is intuitive to imagine extremes in F_{st} reflect historic changes in allele frequencies in response to selection. The problem is that these outlier regions could also reflect changes in allele frequencies due to drift, especially in the context of domestication bottlenecks from ancestral sources, regions of low recombination (and therefore susceptible to linkage drag), or simply regions of high/low absolute diversity between original tropical and temperate material. I'm not an expert in population genomics, but my sense is that problems with F_{st} outlier approaches have been acknowledged for years. Moreover, there will always be outliers in any statistics and only some of these will be the result of selection. Most genome scans of selection use a diversity of approaches and often compare outliers to patterns expected under particular null models – what is the distribution of F_{st} under a neutral or demographic model? It may be that many of the F_{st} regions identified are related to response to strong artificial selection – however, it's hard for the reader to know how many are false positives, due to other evolutionary forces (drift, low diversity or recombination), or an outcome of ascertainment bias resulting from the sampled material or other thresholding. The directionality is also unclear – in what cases are differences due to selection in temperate lines, in tropical lines, or in both? F_{st} may be an especially misleading measure if much of the response to modern selection arose from standing variation? If so, there may be allele freq differences in one breeding scenario along with residual standing variation in the other with little F_{st} differentiation. How robust are the results to the use of other approaches for identifying outliers? Or other thresholds for selection? Have there been other selection scans for domestication genes in maize? Do the results match the regions identified in other studies? One idea is that regions under strong selection should result in reduced variation along a linked block – to what extent are F_{st} differentiated regions associated with extended haplotypes versus small blocks of SNPs?

The remainder of the paper centers on conducting GWAS analyses of two metrics of stability based on regression slopes and MSE. I liked this approach. I've seen few studies conducting genomewide association mapping in the context of GxE and so this is a novel contribution. The authors then ask the degree to which GWAS hits for stability are associated with Fst outlier regions. Overall, they find that putative selected regions explain less variation in yield GxE than unselected regions, suggesting breeding efforts may have reduced GxE.

I have a couple of general questions about the statistical genetic analyses. The majority of the procedures are applications of linear mixed models. These approaches make assumptions about homogeneity of variances and normality of residuals. As such, scale of the data can influence variance partitions and notions of interaction and non-additivity. How were the phenotype distributed? Are the GxE partitionings robust to transformations? To site to site variability?

Given that the plant material is hybrid, with extensive heterozygosity, can the GWAS analyses be completed with dominance and dominance * E terms? It may be that GxE is more pervasive than observed but associated with dominance responses across environments. I can imagine a number of molecular mechanisms where this genetic architecture could be anticipated – for example, environmentally response cis-promoter elements that have been lost/gained in different breeding material.

Page 6, line 127. The sentence lists possible mechanisms of GxE. This sentence is a bit odd, as the simple interpretation of GxE for quantitative traits is that the additive effects of a QTL changes magnitude or sign with the environment. There is no reason to invoke epistasis, dominance, pleiotropy etc.....

It wasn't clear to me how the collected weather data was used in any particular analyses. If not a center feature, then best to move to a supplement.

Overall, this is an interesting set of conceptual ideas and an important dataset. My only reservation is whether the researchers have robustly identified the regions that responded to selection in the context of the panel studied in the G2F project. If so, then the pattern observed and the interpretations presented are a nice contribution.

Reviewer #3 (Remarks to the Author):

To investigate the effect of artificial selection on phenotypic plasticity during tropical to temperate adaptation in maize, Gage et al. phenotyped several hundreds of maize hybrids across 21 environments in North America, identified heavily selected regions by calculating Fst

between temperate and tropical maize, calculated the variability of yield G×E of selected and unselected SNPs and performed GWAS analysis for slope and MSE of hybrid phenotypic plasticity. They concluded that selected regions explained less variability of yield G×E, which implied that breeding efforts for yield traits may reduce the ability to adapt to novel environments. Also, trait-associated SNPs for type II phenotypic plasticity significantly enriched at 5kb upstream of genes which implies that regulatory elements have a predominate role in controlling phenotypic plasticity. The results are novel. While, we do have some general concerns:

For the GWAS analysis with slope and MSE plasticity, the author simply analyzed distribution of associated SNPs without a detailed annotation of associated loci, such as the genes responsible for the plasticity of a specific trait. Also, the authors performed GWAS for the trait per se, the slope and MSE, it will be meaningful to have a comparison to identify shared and specific loci rather than a global comparison of genome distribution of associated SNPs. Further, what's the percentage of plasticity associated SNPs that overlapped with selected regions during temperate maize adaptation?

Below are several specific comments:

- 1.It would be better to have a summary of the experimental design and data acquisition in the Result rather than the Method part since the novel population design of G2F project is so crucial for this paper.
- 2.Line 387,23 and 25 days for days to anthesis and silk, respectively. While in Supplementary Figure 4, they were days to pollen and silk. You should stay consistent.
- 3.In the legend of Figure 1, there were two mistakes for coordinate 1 (.05 should be 0.5). A further question regarding to Figure 1, since the coordinate 1 already classified maize into temperate and tropical groups, what's the reason that the authors choose the temperate and tropical population with coordinate 2 > 0?
- 4.Line 424~427, it's hard to understand that the majority of high Fst SNPs were selected in temperate materials but not tropical ones.
- 5.Line 429~432, SNPs with low Fst tended to have a lower MAF than SNPs with high Fst according to Figure 2. For the concern that SNPs with different allele frequency may influence the variant estimation, the authors should do some simulation for SNPs with comparable Fst and MAF.
- 6.According to the Supplementary 1~3, the power of GWAS for trait per se was stronger than the slope and MSE according the QQ-plot, how to interpret this?
- 7.At the end of chromosome 6, there is a shared peak for the slope of ear height and plant height. It is possible that some master regulators such as transcription factors may control the plasticity of diverse phenotypes. It is valuable to identify shared loci among GWAS results of slope of 5 phenotypes.

Reviewer #1 (Remarks to the Author):

This paper deals with important questions about the genes that control plastic responses of plants to their environments. The authors conclude that alleles that were selected based on yield in temperate conditions (high F_{st} SNPs) differ from unselected SNPs (low F_{st}) for their level of genotype-environment interaction. The analyses indicate a significant reduction in GxE due to selected SNPs. This result is biologically plausible, interesting, and potentially important. Nevertheless, I have some substantial concerns.

Thank you for the interest and useful comments and suggestions on our work.

1) In order to argue that selection explains the plasticity differences between these two classes of polymorphisms, it is important to examine whether the high F_{st} and low F_{st} SNPs differ in ways that might be confounded with F_{st} . For example, do these two groups differ in distance from genes, distance from each other, LD among SNPs, proportions of imputed sites, or other confounding factors?

We agree this is an important consideration. The high and low F_{st} SNPs show very similar distributions for distance to the nearest gene and proportion of missing hybrid genotypes. We do see different distributions for LD among SNPs and distance between SNPs, with higher LD and lower distances between SNPs within the high F_{st} group. This is to be expected, however, as differential selection between the tropical and temperate inbreds used to calculate F_{st} is likely to have acted on regions rather than on randomly distributed single-SNP loci. We have added text on line 463 describing these findings.

2) I am skeptical about some of the methodologies used in this study, especially the reliance on SNPs that have very high missing rates, and using trait information from genotypes with 80% missing data. (See Data Quality, below.)

We have amended the manuscript (starting on lines 320 and 378) to clarify details about the genotypic data quality. F_{st} calculations were performed using high quality SNP data from whole genome sequencing, as detailed in Bukowski et al. (2016; <https://doi.org/10.1101/026963>).

The proportion missing data on a SNP basis for the variance estimation experiment ranges from 0.00 to 0.56 with a mean of 0.16. Proportion missing data on a genotype basis ranges from 0.03 to 0.56 with a mean of 0.16. We changed the text on line 320 to state this.

The proportion missing data for the GWAS ranges from 0.00 to 0.80 with a mean of 0.20 and 95% of SNPs missing less than 0.61. The distribution of missing genotypes is heavily skewed, but there are not many SNPs with large quantities of missing data. We have clarified this on line 378.

3) In a large and polymorphic genome, identifying high and low F_{st} SNPs based on groups of 30 inbred individuals seems like a rather small sample size. Furthermore, these 30 genotypes are not fully independent, given that their identity can be as much as 94%. I presume there were constraints in your data set which kept you from using more individuals (otherwise, one would definitely prefer more genotypes). This sounds like a serious concern -- what are the counter arguments?

We chose to use 30 individuals for each group in order to best balance the percentage of missing data between the two groups. When we selected the 30 individuals from each group with the least missing data, the average for both groups was approximately 20% (from 0.02 to 0.40 for temperate and from 0.13 to 0.21 in tropical inbreds). We are confident that 30 individuals in each group is a sufficient size to accurately estimate F_{st} , as previous publications have shown reasonable F_{st} estimations with as little as 11 inbreds (Gore et al., 2011 doi:10.1126/science.1177837).

The 95% identity by state cutoff was used to ensure we did not include duplicate samples. This value was chosen based on distribution of IBS between duplicate samples by Romay et al., 2013 (doi:10.1186/gb-2013-14-6-r55). Because the temperate group is by nature less diverse than the tropical group, the average distance between a single inbred and the other 29 was approximately 0.20 for the temperate group and 0.25 for the tropical group. The lowest (and also very rare) distances within the temperate group were about 0.10, while the tropical were more stable around 0.25. We have added text on line 304-308 to better explain our choice of sample size and identity by state cutoff.

4) Assignment of sets to locations was based on expected maturity (line 198), hence there is a genotype-environment correlation. What is the magnitude of this correlation, and how does this influence these genetic outcomes?

Sets were assigned to locations in an effort to put as many unique hybrids in as many locations as possible while still operating within the constraints of resources and photoperiod sensitivity. Had we grown all hybrids in all locations, we would have seen abnormal phenotypes from attempting to grow individuals in conditions under which they would not normally flower. This may result in overestimated phenotypic variability within hybrid genotypes. We acknowledge the reviewer's concern over the genotype-environment correlation, but believe that under a completely balanced design we would result in incorrectly estimating hybrids' responses to environments due to growing them outside their appropriate set of environments.

5) The sequence and details of analyses is logical, but there is a lot going on, and it is not easy to follow. At the end of the introduction, please provide a brief road-map to your approach.

We have expanded the text, starting on line 174, to summarize our analyses, as suggested. Another reviewer had a similar suggestion, so we have also added a new Figure 1 which provides a flowchart detailing each experiment.

Data quality:

I think the filtering thresholds for including hybrids across environments are much too lax (pages 15 and 16). Across 21 environments, you are only excluding hybrids with fewer than four data points, or fewer than six environments. From this you cannot get a reliable slope, nor can you be certain that you have sampled the range of environmental space, or whether differential slopes might simply be due to sampling different parts of the gradient. For the genotypes used, did you check whether the regressions deviate from linearity? (An interaction of genotype by quadratic effects of the gradient?)

Distributions of R^2 for our regressions were centered on 0.6 to 0.9, depending on the trait, which gives us confidence that these regressions explain a sizable portion of the variation. Given the unbalanced nature of our data (not all hybrids were tested across all locations) we cannot always sample the entire range of the environmental space for each trait. Most hybrids were evaluated in between 5 and 10 environments, which based on previous studies (Finlay & Wilkinson used 7 in their original 1963 publication, doi:10.1071/AR9630742) is adequate to estimate a slope. In addition, we rely on the mean squared error (MSE) of our regressions to capture non-linear response across environments rather than map and categorize results from a number of nonlinear regression terms,

Using SNPs that might have as much as 79% missing data strikes me as excessive (line 315). Please provide numbers and distributions so that we can have a clear understanding of the extent of missing SNP data (across SNPs and across genotypes).

As described in response to remark #2, above, we have added text on lines 320 and 378 detailing the amount of missing data on a per-SNP and per-hybrid basis.

Please provide information on the success of imputation. In particular, for SNPs that were called with high confidence, what is the success rate when you hide this information and impute these SNPs?

The imputation accuracy rate for FILLIN was documented in Swarts et al. (2014; doi:10.3835/plantgenome2014.05.0023) as having an accuracy of 0.996 when imputing among temperate inbreds which represents the set of germplasm genotyped for this study. We added text on line 283 to more specifically describe the imputation accuracy. Approximately 16% of the synthetic hybrid genotypes were not imputed by FILLIN, and needed to be imputed for variance estimations. Those few sites were imputed by random draws weighted by the allele frequencies at each locus. To test the accuracy of this second imputation step, we ran 10,000 iterations of masking a single SNP call and imputing. The imputation accuracy was 85.6%, and is now documented on line 324 of the manuscript. It is important to note that the 85.6% imputation accuracy is only relevant for the 16% of missing SNPs that were not imputed by FILLIN, which has a much higher imputation accuracy.

For linkage disequilibrium calculations (line 360), what proportion of the data have non-missing data for both SNPs? What is the mean number of non-missing SNP pairs that was used for linkage disequilibrium calculations?

The mean number of non-missing SNP pairs for LD calculations was 404 and ranged from 75 to 552.

Minor points:

For those of us who don't work on maize, it would be helpful to have more background about the different populations, their histories, and how they differ.

Thank you for pointing this out. We have added text on line 201 in the methods providing more information about the eight pools of inbreds used to create hybrids.

It is not clear that these eight groups should be described as heterotic groups (line 185). What is the

evidence that they are heterotic? Also, heterosis is a characteristic of a *cross* between different parents, how can they be called heterotic unless you specify a comparison between groups?

Some of the pools we used are often referred to as heterotic groups within the maize community, though we acknowledge the reviewer's concern that we have not provided any information to demonstrate that they are in fact heterotic. We have dropped the "heterotic" description from the manuscript.

"We observed reduced genetic variance attributable to high Fst SNPs compared to low Fst SNPs for both grain yield and plant height, but high Fst genetic effects still accounted for 11.2% (grain yield)" -- We also need to know how this 11.2% figure compares to the genetic variance of grain yield explained by low Fst SNPs.

Thank you for this suggestion. We have expanded this section (line 470) to include a statement providing more details about the proportion of the genetic variance explained by low Fst SNPs.

The meaning of "set" (a pool-by-tester family) is not clear from the written description. I understand that a set has a group of female parents consisting of a given "heterotic" pool. Does this set have a single male tester (such as PB80), or does a given set have five male parents (PB80 through LH185)? In the total experiment, what is the number of sets, and how many full sib-ships are there? What is the total sample size of the experiment analyzed here?

The pools and male testers used to make each set are detailed in Supplemental Table 1 which shows that there are 30 sets, each having a single male tester. There are no full sibs, only half sib progeny. The total number of unique hybrids is 858 (line 188), and the methods have been expanded to include the total number of individual plots (12,678; line 189) as suggested here.

The different results between type II and type III stability (lines 505-509) may not reflect a biological difference. Estimation of slope for type II stability provides a smoothing affect across different environments, whereas stochastic differences among different environments will inflate type III estimates of variability. Thus, experimental noise may explain the lack of relationship between regulatory elements and type III stability.

We agree with the reviewer and have added text on line 606 to offer this perspective as an alternative reason for different results between types II and III.

"populations with a greater capacity for phenotypic plasticity have been shown to have greater fitness than populations that are less able to respond to their environment" (line 121) -- On the other hand, other examples find that plasticity may be non-adaptive (e.g., Ghalambor, et al, 2015, Non-adaptive plasticity potentiates rapid adaptive evolution of gene expression in nature. Nature 525: 372-375).

Thank you for the suggestion. We have added text to the introduction on line 122 to include non-adaptive plasticity.

For the MDS, What proportion of the variance is explained by the first two axes? Is this sufficient to identify the two groups of 30 genotypes?

For our MDS, we use a nonmetric ordinal technique where there are no hidden axes of variation linked to dispersion parameters (variance) in any order. The implemented method is based instead on nonparametric assumptions. Unlike most ordination methods (based on PCA/SVD) where the optimum number of components is selected based on the cumulative proportion of variance explained by the first n components, in this case the number of dimensions to keep is chosen based on a stress statistic. The stress statistic resembles the goodness of fit (controlling model fit and model complexity) between ordination-based distances and the distances predicted by regression using a given number of components. Alternatively, it can be viewed as the proportion of variability *not* explained with the current number of components. In our case, the corresponding values using 1, 2, 3 and 4 components were 0.38, 0.25, 0.20 and 0.15. Quinn and Keough (2002) have suggested using the number of components that would ensure a stress value smaller than 0.3. In our case, that level was achieved with the first 2 components.

You suggest (line 536) that selection against deleterious alleles may explain the higher stability of selected SNPs. Does this imply that they are unconditionally deleterious? This is very different than constitutively deleterious alleles.

In this part of the discussion, we were referring to alleles that have been selected against as deleterious in the context of temperate breeding programs. We have modified the text on line 574 to make that clearer.

What factors might explain the difference in results between Wallace et al. vs. the current study?

The Wallace et al. study had more power than this study due to the much larger number of phenotypes used, which we believe may have led to the stronger genic signal observed in his results. In addition to having fewer phenotypes we are further dividing our intragenic and gene-proximal bins into upstream and downstream categories, which also reduces power relative to his study but provides a more accurate description of the biological effect.

Line 353: What is a “non-monomorphic hybrid genotype”? Here, is genotype synonymous with SNP?

Thank you for your comment. The wording of that phrase was ambiguous. We have edited it for clarity (line 377).

Reviewer #2 (Remarks to the Author):

Gage and colleagues present results of a large-scale collaborative field trial, population genomic analysis, and GWAS mapping in maize to explore the impact of domestication on gene by environment interaction (GxE). GxE plays an important role in plant growth and performance and the study frames the question of how artificial selection during domestication affects standing GxE variation for agronomic traits. The implications of the study are broad, important and potentially provide insight into the mechanisms leading to the evolution of broadly stable versus regionally adapted cultivars.

The projects leverages an exciting new collaboration, the G2F project, wherein a common set of maize

hybrids and checks are grown at many locations across North America. The material was phenotyped for a core set of agronomic traits ranging from flowering time to grain yield and the plant material genotyped using a combination of resequencing, GBS, and imputation. These data are analyzed using statistical genetic approaches to partition sources of variation into E, G, GxE, and design factors. This multi-location trial is novel in that the plant material was picked to broadly represent maize diversity, tests effects in hybrids, and can be linked by association to SNP variation.

Thank you for the summary and for your helpful suggestions on our manuscript.

One of the main goals for the study is to ask whether genomic regions under selection during domestication affects variance in GxE. The authors attempt to identify these regions by comparing SNP variation in a chosen set of 30 temperate and 30 tropical adapted maize inbreds. This material would have been the result of strong selection during breeding programs and the authors aim to identify the regions which responded to selection by identifying outliers in the relative differentiation metric F_{st} .

F_{st} measures the degree of among group versus within group allelic variation. It is intuitive to imagine extremes in F_{st} reflect historic changes in allele frequencies in response to selection. The problem is that these outlier regions could also reflect changes in allele frequencies due to drift, especially in the context of domestication bottlenecks from ancestral sources, regions of low recombination (and therefore susceptible to linkage drag), or simply regions of high/low absolute diversity between original tropical and temperate material. I'm not an expert in population genomics, but my sense is that problems with F_{st} outlier approaches have been acknowledged for years. Moreover, there will always be outliers in any statistics and only some of these will be the result of selection. Most genome scans of selection use a diversity of approaches and often compare outliers to patterns expected under particular null models – what is the distribution of F_{st} under a neutral or demographic model? It may be that many of the F_{st} regions identified are related to response to strong artificial selection – however, it's hard for the reader to know how many are false positives, due to other evolutionary forces (drift, low diversity or recombination), or an outcome of ascertainment bias resulting from the sampled material or other thresholding. The directionality is also unclear – in what cases are differences due to selection in temperate lines, in tropical lines, or in both? F_{st} may be an especially misleading measure if much of the response to modern selection arose from standing variation? If so, there may be allele freq differences in one breeding scenario along with residual standing variation in the other with little F_{st} differentiation. How robust are the results to the use of other approaches for identifying outliers? Or other thresholds for selection? Have there been other selection scans for domestication genes in maize? Do the results match the regions identified in other studies? One idea is that regions under strong selection should result in reduced variation along a linked block – to what extent are F_{st} differentiated regions associated with extended haplotypes versus small blocks of SNPs?

We acknowledge the reviewer's concern with using \$F_{st}\$ as a statistic for identifying differentially selected regions. We propose that regions identified as having high \$F_{st}\$ are more likely to have undergone selection, and follow this with further investigation of the role those regions play in GxE. The utility of \$F_{st}\$ in this study is not to identify regions that have certainly undergone selection, but rather to identify regions that are simply more likely to have been selected. Creating a null distribution for any selection statistic under a neutral or demographic model is subject to numerous problems, one of which is that

simulations are often based on overly simplistic models (Walsh 2008, doi: 10.1007/s10681-007-9465-8). The history of temperate and tropical maize breeding is complex and would be exceedingly difficult to model accurately. We instead follow one of the suggestions from the same review mentioned above (Walsh 2008), which proposes using a large number of markers and identifying outliers as selection candidates. We have modified the text throughout the manuscript to clarify that we are only identifying selection candidates, not definitively selected regions. We have tried to acknowledge and address some of the reviewer's concerns such as: directionality of selection (line 421-427 of the original manuscript); and whether allele frequencies vary between temperate and tropical lines or between high and low F_{st} SNPs (Figures 2a,c). There are a number of other methods for identifying signals of selection, but just like F_{st} each comes with its own set of drawbacks. This makes identification of a single best statistic for identifying selection candidates unlikely. Again, we stress that we are not conducting a scan for selection, only identifying regions that are more likely to have been selected for use in further analysis (GxE variance models).

While there have been previous scans identifying domestication and improvement genes (e.g., Hufford 2012, and numerous papers from the Doebley lab), the tropical materials we used are neither undomesticated nor unimproved so we believe our F_{st} calculations reflect instead candidates of selection for adaptation to modern temperate agriculture. We therefore do not expect overlap between the high F_{st} regions we identify and regions selected during domestication. We have calculated LD and distance between SNPs within the high and low F_{st} SNP sets. We found that the high F_{st} SNPs have higher LD and tend to be closer to each other than the low F_{st} SNPs do, which is to be expected because selection would act on haplotypes or genomic regions rather than on individual SNPs, creating groups of linked high F_{st} SNPs. This provides some evidence for selection on extended haplotypes, as the reviewer mentions. Higher LD and smaller distances between the high F_{st} SNPs also provide evidence against the concern that our high F_{st} SNPs could be largely false positives – we would expect false positives to be more randomly distributed across the genome.

The remainder of the paper centers on conducting GWAS analyses of two metrics of stability based on regression slopes and MSE. I liked this approach. I've seen few studies conducting genomewide association mapping in the context of GxE and so this is a novel contribution. The authors then ask the degree to which GWAS hits for stability are associated with F_{st} outlier regions. Overall, they find that putative selected regions explain less variation in yield GxE than unselected regions, suggesting breeding efforts may have reduced GxE.

The reviewer describes the major points of the experimental workflow correctly, though we did not address the overlap between GWAS hits for stability and F_{st} outlier regions. We did look for overlap between the stability-associated SNPs and the high F_{st} regions, but found no major colocalization. We have also added a new Figure 1 which provides a flowchart of the experiments from this study, and expanded the summary of the experimental workflow starting on line 174.

I have a couple of general questions about the statistical genetic analyses. The majority of the procedures are applications of linear mixed models. These approaches make assumptions about homogeneity of variances and normality of residuals. As such, scale of the data can influence variance partitions and notions of interaction and non-additivity. How were the phenotype distributed? Are the GxE partitionings robust to transformations? To site to site variability?

This is an important consideration and one that we considered carefully. In general terms, the required assumptions for ANOVA (e.g., normality on the errors, homoscedasticity of variances for environments) are typically not met when working with multi-environment data due to different dispersion patterns of genotypes (direction and extension) in different environments (GxE interaction). Specifically with regards to our study, any level of concern is somewhat lessened given that we did not attempt to make inferences (test hypothesis or contrasts, confident intervals) on the location parameters, and so meeting all the ANOVA assumptions is not strictly required.

Still, to help address this reviewer's comment, even though our raw phenotypic values did not follow normal or bell-shaped distributions, the assumption of normality is expected to be made on the residuals since the inclusion of other factors will help to explain an important proportion of the variability. For our 1,000 replicated model fittings, the residual vectors were computed and averaged across replicates. The histogram of the mean residuals showed a distribution that approaches normality (perhaps up to a constant).

We did not use any formal tests for homogenous variability because those tests frequently assume normality, which our phenotypes per se and residuals do not strictly meet. Some authors (e.g., Dean and Voss, "Design and analysis of experiments", 1999) have suggested that if the ratio between the estimated variance of the treatments (in this case, environments) with the largest and the smallest variability does not exceed the value of five, the assumption of homoscedasticity is likely satisfied. In our grain yield models, we see a ratio of 3.3 between the environments with the most and least variability. In our plant height models we see a ratio of 5.2, which is slightly above the proposed threshold. We do not see a clear separation of variance components for high and low F_{st} regions in the plant height models, and therefore are less concerned about validation of normality assumptions than for the grain yield models.

Finally, we did not consider it necessary perform any of the common transformations on the data to meet ANOVA assumptions since these transformations do not preserve data scaling in the estimation process, preventing us from providing accurate values of the variance components in data scale.

Given that the plant material is hybrid, with extensive heterozygosity, can the GWAS analyses be completed with dominance and dominance * E terms? It may be that GxE is more pervasive than observed but associated with dominance responses across environments. I can imagine a number of molecular mechanisms where this genetic architecture could be anticipated – for example, environmentally response cis-promoter elements that have been lost/gained in different breeding material.

We think that the reviewer presents a very good suggestion here, but we feel that our experimental design is not appropriate to test these questions. The hybrids used for this study (checks aside) were generated with only five male testers and only a fraction of those were typically tested in specific locations.

Page 6, line 127. The sentence lists possible mechanisms of GxE. This sentence is a bit odd, as the simple interpretation of GxE for quantitative traits is that the additive effects of a QTL changes magnitude or sign with the environment. There is no reason to invoke epistasis, dominance, pleiotropy etc.....

We agree with the reviewer about the simple interpretation of GxE. The review cited was referring to mechanisms affecting GxE, such as observing different changes in QTL effects dependent on, e.g., genotypes at multiple loci, traits considered, or number of unique alleles at a locus. We have edited the

text on line 129 to make this clearer.

It wasn't clear to me how the collected weather data was used in any particular analyses. If not a center feature, then best to move to a supplement.

We have moved the text regarding weather data collection to a supplement, as suggested.

Overall, this is an interesting set of conceptual ideas and an important dataset. My only reservation is whether the researchers have robustly identified the regions that responded to selection in the context of the panel studied in the G2F project. If so, then the pattern observed and the interpretations presented are a nice contribution.

Reviewer #3 (Remarks to the Author):

To investigate the effect of artificial selection on phenotypic plasticity during tropical to temperate adaptation in maize, Gage et al. phenotyped several hundreds of maize hybrids across 21 environments in North America, identified heavily selected regions by calculating F_{st} between temperate and tropical maize, calculated the variability of yield $G \times E$ of selected and unselected SNPs and performed GWAS analysis for slope and MSE of hybrid phenotypic plasticity. They concluded that selected regions explained less variability of yield $G \times E$, which implied that breeding efforts for yield traits may reduce the ability to adapt to novel environments. Also, trait-associated SNPs for type II phenotypic plasticity significantly enriched at 5kb upstream of genes which implies that regulatory elements have a predominate role in controlling phenotypic plasticity. The results are novel. While, we do have some general concerns:

Thank you to the reviewer for taking the time to read and review the paper, and for the accurate summary.

For the GWAS analysis with slope and MSE plasticity, the author simply analyzed distribution of associated SNPs without a detailed annotation of associated loci, such as the genes responsible for the plasticity of a specific trait. Also, the authors performed GWAS for the trait per se, the slope and MSE, it will be meaningful to have a comparison to identify shared and specific loci rather than a global comparison of genome distribution of associated SNPs. Further, what's the percentage of plasticity associated SNPs that overlapped with selected regions during temperate maize adaptation?

Because the hypothesis we were testing was stated with regards to overarching trends in SNP locations, we did not annotate specific genes underneath the SNPs we identified. Our focus is to test the hypothesis of enrichment of associations in regulatory regions over other regions and not on individual gene discovery. We would encourage future work investigating the genomic regions tagged by our GWAS, but feel this could be the subject of an entirely separate study. We have created Supplemental figure 10, which shows the location of the 250 SNPs chosen for traits per se, slope, and MSE. The figure illustrates that there is no systematic overlap between loci identified as associated with slope, MSE, and traits per se. We have also compared the list of SNPs associated with stability to the high F_{st} SNPs and found no major

colocalization.

Below are several specific comments:

1.It would be better to have a summary of the experimental design and data acquisition in the Result rather than the Method part since the novel population design of G2F project is so crucial for this paper.

We have added text on line 410 providing a concise summary of the experiment and data acquisition as suggested. The detailed explanation of the experimental design and data acquisition remains in the Methods.

2.Line 387,23 and 25 days for days to anthesis and silk, respectively. While in Supplementary Figure 4, they were days to pollen and silk. You should stay consistent.

Thanks to the reviewer for catching this. We have changed Supplementary figure 4 to be consistent with the manuscript.

3.In the legend of Figure 1, there were two mistakes for coordinate 1 (.05 should be 0.5). A further question regarding to Figure 1, since the coordinate 1 already classified maize into temperate and tropical groups, what's the reason that the authors choose the temperate and tropical population with coordinate 2 > 0?

Again, much thanks to the reviewer for noticing this mistake. We have changed the legend of Figure 1. Within the MDS coordinate 1 alone is not enough to distinguish the temperate and tropical groups. Tropical lines are mostly confined to the material within the green box – there are individuals with coordinate 1 > 0.5 and coordinate 2 < 0 that are not tropical.

4.Line 424~427, it's hard to understand that the majority of high Fst SNPs were selected in temperate materials but not tropical ones.

Thank you for pointing this out. We have edited the text (line 455) for clarity.

5.Line 429~432, SNPs with low Fst tended to have a lower MAF than SNPs with high Fst according to Figure 2. For the concern that SNPs with different allele frequency may influence the variant estimation, the authors should do some simulation for SNPs with comparable Fst and MAF.

The low Fst SNPs do tend to have slightly lower MAF than high Fst SNPs. SNPs with lower MAF will contribute less variance than SNPs with intermediate MAF. Our results show low Fst SNPs contributing more GxE variance for grain yield, whereas if the amount of GxE variance was only a function of allele frequency, we would expect the low Fst SNPs to contribute less GxE variance. In this regard, we feel that the results are rendered, if anything, somewhat conservative by the differences between MAF distributions.

6. According to the Supplementary 1~3, the power of GWAS for trait per se was stronger than the slope and MSE according to the QQ-plot, how to interpret this?

The slope and MSE are both derived traits, whereas the traits per se were measured directly. This contributes additional noise to the slope and MSE estimations, which may have reduced our power to detect signal for them.

7. At the end of chromosome 6, there is a shared peak for the slope of ear height and plant height. It is possible that some master regulators such as transcription factors may control the plasticity of diverse phenotypes. It is valuable to identify shared loci among GWAS results of slope of 5 phenotypes.

We have added Supplemental figure 10, which shows colocalization of SNPs associated with traits per se, slope, and MSE. Though there are a small number of regions with shared loci among traits that could be the target of further investigation, no systematic overlapping is observed in associated regions.

Reviewers' comments:

Reviewer #1 (Remarks to the Author):

This paper deals with fundamental biological questions in a very important study system. The experimental data set is remarkable, and the analyses are excellent. The responses and revisions are very good. I have one major comment, and a few minor ones.

Major point regarding divergence and F_{st} :

On page 6 of the rebuttal, Reviewer 2 raises an important point about F_{st} . One goal of this paper is to “identify regions that show high divergence in allele frequency” (MS line 175), and F_{st} is used for this purpose. However, F_{st} is a *relative* measure of divergence between populations, which may give different results than *absolute* measures of divergence (Cruickshank & Hahn, Molecular Ecology, 2014). C&H reanalyzed previously published studies of species pairs, and found that high F_{st} regions did not show elevated allele frequency divergence. Instead, high F_{st} regions were due to reduced diversity rather than increased divergence. They conclude that previous inferences of local divergence are artifactual, and that F_{st} is the wrong measure to use for such comparisons.

On page 7 of the rebuttal, the authors discuss different estimators of divergence. While those arguments are plausible, nevertheless, F_{st} has limitations, and it is important to verify that the apparent biological conclusions are not an artifact of a particular estimator (as was the case in the C&H reanalysis). If the same pattern is found for both relative and absolute measures of divergence, this will strengthen our confidence in these conclusions. (I hate to add hurdles to the process now, but it better than possible unpleasant surprises later.)

Minor points:

MS Line 307: There are two different sample size issues for F_{st} . Ten or thirty individuals per group is fine to calculate mean F_{st} across thousands of SNPs, but this low number of individuals will add considerable noise to identification of high and low F_{st} SNPs. Please add a sentence acknowledging this point.

Rebuttal, Reviewer 1 Point 3: An estimate of F_{st} at a particular locus depends on the number of individuals genotyped at that locus. (Missing data increases F_{st} , just as a population bottleneck does.) Do the high F_{st} SNPs have data from fewer individuals?

Rebuttal, Reviewer 1 Point 4: I understand your point, and I'm not suggesting that you should

have done this differently. Still, a genotype-environment correlation does exist in your data. What can you say about the consequences of this correlation?

Rebuttal, Reviewer 1, Page 3, Imputation: This response is convincing.

Reviewer #2 (Remarks to the Author):

Gage et al. present results from genomic analyses and field trials exploring gene-by-environment interaction in domesticated maize. The study centers on inferring candidate genomic regions that were putatively artificially selected or neutral during domestication and asking the degree to which these regions explain GxE in field trials. The authors argue that genomic regions harboring allele frequency changes associated with artificial selection contribute less to GxE for yield than neutral regions. The interpretation is that breeders indirectly selected for stable yield during domestication. The analyses also suggest that regions associated with stability are enriched in promoters. Overall, I think this is a novel and interesting project. I like the linking of genomic inference with field trials and more studies of plasticity and GxE are needed - the paper touches on both interesting biology as well as important phenomena in applied plant breeding.

However, I'm still concerned whether the inference of selection is robust. As noted by the authors, *Fst* divergence between two groups could have been driven by a number of processes, including selection and drift (lines 452-458). The authors acknowledge that their conclusions are contingent on the assumption that selection in the tropics and drift are rare. The authors really provide no evidence favoring one mechanism over another and simply rest on an argument that strong selection "is expected in the temperate material" (line 458). However, I think it would be equally robust to argue that strong drift in the domestic bottleneck is also expected. Surely, the high *Fst* regions are a mixture of these processes. I was hoping that additional analyses could help parse some of these alternatives. For example, a careful exploration of nucleotide diversity could potentially sort out divergent versus directional selection in the tropics or temperate regions. As it stands, the main result of the paper is based on weak inference. If additional lines of evidence cannot be provided in favor of selection, I think the authors need to more carefully word some of their interpretation and perhaps tone down the title and abstract. If nothing else, the authors need to include more discussions of the caveats and strength of their interpretation.

My original review also asked about the distribution of data and statistical testing and inference. This is a thorny issue as well. It's not uncommon for researchers to work on transformed scales to minimize non-additive (GxE) patterns and to better meet the assumptions of parametric testing. Nevertheless, there are some benefits of working on the raw scale. It would help been helpful to see how robust the results and interpretation are in terms of obvious transformations.

Instead, the authors argue that its probably not an issue in their data. At some level, the reader just needs to trust the team that this is so.....but it seems like it would be reasonable to mention some of these issues in the methods section.

The authors make a number of editorial changes that improve the manuscript, including clarifying a number of features of the experimental approach and workflow.

Reviewer #3 (Remarks to the Author):

The authors have made adequate editorial changes. But I am still a little bit worried about the representativeness of temperate lines in the blue box with respect to the new Figure 2 since Oh43 and W22 are two classical temperate inbred lines which were outside the blue box. How many genetic variance among the inbred lines that can be explained by coordinate1 alone? Actually, the density in Figure 2 was biased towards the region with Coordinate2 ≥ 0 which may push the authors to set up the criteria with Coordinate2 ≥ 0 . While, the authors should be aware of the possibility that different set of temperate lines can end up with the identification of largely different selective regions.

Reviewer #1 (Remarks to the Author):

This paper deals with fundamental biological questions in a very important study system. The experimental data set is remarkable, and the analyses are excellent. The responses and revisions are very good. I have one major comment, and a few minor ones.

Major point regarding divergence and F_{st} :

On page 6 of the rebuttal, Reviewer 2 raises an important point about F_{st} . One goal of this paper is to “identify regions that show high divergence in allele frequency” (MS line 175), and F_{st} is used for this purpose. However, F_{st} is a *relative* measure of divergence between populations, which may give different results than *absolute* measures of divergence (Cruickshank & Hahn, Molecular Ecology, 2014). C&H reanalyzed previously published studies of species pairs, and found that high F_{st} regions did not show elevated allele frequency divergence. Instead, high F_{st} regions were due to reduced diversity rather than increased divergence. They conclude that previous inferences of local divergence are artifactual, and that F_{st} is the wrong measure to use for such comparisons.

On page 7 of the rebuttal, the authors discuss different estimators of divergence. While those arguments are plausible, nevertheless, F_{st} has limitations, and it is important to verify that the apparent biological conclusions are not an artifact of a particular estimator (as was the case in the C&H reanalysis). If the same pattern is found for both relative and absolute measures of divergence, this will strengthen our confidence in these conclusions. (I hate to add hurdles to the process now, but it better than possible unpleasant surprises later.)

Thank you for your suggestion of considering the Cruickshank & Hahn paper. This work provides a very useful perspective and it has helped thinking about our experiment from the perspective of their work. Their work suggests that absolute measures of divergence are preferable to relative measures in experiments dealing with species (or subpopulations) that are experiencing significant gene flow between them. In such cases, absolute measures of divergence are a better metric for identifying genomic regions that are resistant to gene flow.

Our case, however, does not fit that model. Although it is expected that some level of gene flow might occur between our two subpopulations, the effect of such phenomenon is expected to be quite small compared to the effect of divergent selection for performance in different conditions (i.e.: temperate vs tropical). The scenario present in our study is more closely aligned with the “alternative model” that Cruickshank & Hahn discuss near the end of the paper. In this alternative model, genomic regions of relative divergence are due to differential adaptation or selection. In scenarios characterized by differential selection, the authors state that D_{xy} can be equivalent between neutral and selected loci (or even lower in selected loci, in some cases).

Even though the particular statistic suggested by Cruickshank & Hahn is not expected to show differences in our particular scenario, we recognize that F_{st} alone is not sufficient to identify selected regions with complete confidence. We therefore have also assessed per-site nucleotide diversity in the

putatively selected (high F_{st}) and unselected (low F_{st}) regions to establish a more absolute measure of divergence. Our analysis found substantially lower diversity in the putatively selected regions compared to unselected regions, which contributes to their relative divergence and can be due to selection. The median level of nucleotide diversity in the putatively selected regions is similar to the median level of nucleotide diversity observed in known selection candidates previously published by Gore et al. 2009. We have added these results in Supplemental Figure 8 and included relevant text on lines 297 and 442. We have also added text on line 547 to better address our assumptions related to using F_{st} .

Minor points:

MS Line 307: There are two different sample size issues for F_{st} . Ten or thirty individuals per group is fine to calculate mean F_{st} across thousands of SNPs, but this low number of individuals will add considerable noise to identification of high and low F_{st} SNPs. Please add a sentence acknowledging this point.

We've added text in the discussion on line 553 mentioning that the F_{st} data are noisy due to sample size issues.

Rebuttal, Reviewer 1 Point 3: An estimate of F_{st} at a particular locus depends on the number of individuals genotyped at that locus. (Missing data increases F_{st} , just as a population bottleneck does.) Do the high F_{st} SNPs have data from fewer individuals?

Thanks for your suggestion to look into this. The vast majority of SNPs in the high F_{st} regions had less than 10% missing data in the Tropical lines and less than 30% missing data in the Temperate lines. The distributions of missing data across the high F_{st} SNPs are similar to the distributions of missing data across all SNPs.

Rebuttal, Reviewer 1 Point 4: I understand your point, and I'm not suggesting that you should have done this differently. Still, a genotype-environment correlation does exist in your data. What can you say about the consequences of this correlation?

We have added text on line 589 acknowledging that such correlation and describing the consequences that it might have in terms of the assumptions that accompany such experimental limitation.

Rebuttal, Reviewer 1, Page 3, Imputation: This response is convincing.

Reviewer #2 (Remarks to the Author):

Gage et al. present results from genomic analyses and field trials exploring gene-by-environment interaction in domesticated maize. The study centers on inferring candidate genomic regions that were

putatively artificially selected or neutral during domestication and asking the degree to which these regions explain GxE in field trials. The authors argue that genomic regions harboring allele frequency changes associated with artificial selection contribute less to GxE for yield than neutral regions. The interpretation is that breeders indirectly selected for stable yield during domestication. The analyses also suggest that regions associated with stability are enriched in promoters. Overall, I think this is a novel and interesting project. I like the linking of genomic inference with field trials and more studies of plasticity and GxE are needed - the paper touches on both interesting biology as well as important phenomena in applied plant breeding.

However, I'm still concerned whether the inference of selection is robust. As noted by the authors, F_{st} divergence between two groups could have been driven by a number of processes, including selection and drift (lines 452-458). The authors acknowledge that their conclusions are contingent on the assumption that selection in the tropics and drift are rare. The authors really provide no evidence favoring one mechanism over another and simply rest on an argument that strong selection "is expected in the temperate material" (line 458). However, I think it would be equally robust to argue that strong drift in the domestic bottleneck is also expected. Surely, the high F_{st} regions are a mixture of these processes. I was hoping that additional analyses could help parse some of these alternatives. For example, a careful exploration of nucleotide diversity could potentially sort out divergent versus directional selection in the tropics or temperate regions. As it stands, the main result of the paper is based on weak inference. If additional lines of evidence cannot be provided in favor of selection, I think the authors need to more carefully word some of their interpretation and perhaps tone down the title and abstract. If nothing else, the authors need to include more discussions of the caveats and strength of their interpretation.

The reviewer makes an excellent point and we recognize the importance of this point in terms of the interpretation of results. To help address this important point, we assessed nucleotide diversity in the high and low F_{st} regions of the temperate and tropical inbreds, hypothesizing that in the case of directional selection within either the temperate or tropical lines, the high F_{st} windows would have low nucleotide diversity in only one or the other of the subpopulations. In the case of divergent selection, we expected low nucleotide diversity in the high F_{st} windows in both subpopulations. Most of the high F_{st} windows appear to have reduced nucleotide diversity in both subpopulations, supporting divergent selection. We have added text to the methods (line 297) and results (line 442) detailing nucleotide diversity analysis. We have also added text on line 547 that discusses the assumptions we are making by using F_{st} and incorporates the idea that our interpretation of the results is contingent on the assumption that selection has a greater effect than drift in changing allelic frequencies in this particular situation. We've also carefully reworded sections throughout the manuscript to reflect the fact that our selection candidate regions are only putative,

My original review also asked about the distribution of data and statistical testing and inference. This is a thorny issue as well. It's not uncommon for researchers to work on transformed scales to minimize non-additive (GxE) patterns and to better meet the assumptions of parametric testing. Nevertheless, there are some benefits of working on the raw scale. It would help been helpful to see how robust the

results and interpretation are in terms of obvious transformations. Instead, the authors argue that its probably not an issue in their data. At some level, the reader just needs to trust the team that this is so.....but it seems like it would be reasonable to mention some of these issues in the methods section.

We have added text on line 334 in the methods section to explain that we did evaluate model assumptions and based on assumption results did not explore phenotypic transformation.

The authors make a number of editorial changes that improve the manuscript, including clarifying a number of features of the experimental approach and workflow.

Reviewer #3 (Remarks to the Author):

The authors have made adequate editorial changes. But I am still a little bit worried about the representativeness of temperate lines in the blue box with respect to the new Figure 2 since Oh43 and W22 are two classical temperate inbred lines which were outside the blue box. How many genetic variance among the inbred lines that can be explained by coordinate1 alone? Actually, the density in Figure 2 was biased towards the region with Coordinate2 ≥ 0 which may push the authors to set up the criteria with Coordinate2 ≥ 0 . While, the authors should be aware of the possibility that different set of temperate lines can end up with the identification of largely different selective regions.

Thank you for raising these points. The question about proportion variance explained was also raised by another reviewer in the last set of reviews. I've repeated our response to that question here:

For our MDS, we use a nonmetric ordinal technique where there are no hidden axes of variation linked to dispersion parameters (variance) in any order. The implemented method is based instead on nonparametric assumptions. Unlike most ordination methods (based on PCA/SVD) where the optimum number of components is selected based on the cumulative proportion of variance explained by the first n components, in this case the number of dimensions to keep is chosen based on a stress statistic. The stress statistic resembles the goodness of fit (controlling model fit and model complexity) between ordination-based distances and the distances predicted by regression using a given number of components. Alternatively, it can be viewed as the proportion of variability not explained with the current number of components. In our case, the corresponding values using 1, 2, 3 and 4 components were 0.38, 0.25, 0.20 and 0.15. Quinn and Keough (2002) have suggested using the number of components that would ensure a stress value smaller than 0.3. In our case, that level was achieved with the first 2 components.

We appreciate your concern over identifying different genomic regions when using different sets of temperate and tropical lines, and have added a statement in the discussion (line 555) clarifying this point.

Reviewers' Comments:

Reviewer #1 (Remarks to the Author):

I think the authors have done a good job with their responses and edits. This manuscript deals with important questions, with a large and challenging data set. I think this is a substantial contribution to understanding the evolution of complex traits in domesticated species.

Reviewer #2 put his/her comment in Remarks to Editor section. He think the authors have addressed my concerns or have added sufficient caveats to their interpretation and inference.

Reviewer #3 (Remarks to the Author):

The authors have made appropriate responses to my previous question regarding to the representativeness of temperate and tropical maize accessions.

I have no further questions.

Thank you!